# Synergistic and antagonistic drug interactions in the treatment of systemic fungal infections

Morgan A Wambaugh, Steven T Denham, Magali Ayala, Brianna Brammer, Miekan A Stonhill, Jessica CS Brown*

Division of Microbiology and Immunology, Pathology Department, University of Utah School of Medicine, Salt Lake City, United States

**Abstract** Invasive fungal infections cause 1.6 million deaths annually, primarily in immunocompromised individuals. Mortality rates are as high as 90% due to limited treatments. The azole class antifungal, fluconazole, is widely available and has multi-species activity but only inhibits growth instead of killing fungal cells, necessitating long treatments. To improve treatment, we used our novel high-throughput method, the overlap$^2$ method (O2M) to identify drugs that interact with fluconazole, either increasing or decreasing efficacy. We identified 40 molecules that act synergistically (amplify activity) and 19 molecules that act antagonistically (decrease efficacy) when combined with fluconazole. We found that critical frontline beta-lactam antibiotics antagonize fluconazole activity. A promising fluconazole-synergizing anticholinergic drug, dicyclomine, increases fungal cell permeability and inhibits nutrient intake when combined with fluconazole. In vivo, this combination doubled the time-to-endpoint of mice with *Cryptococcus neoformans* meningitis. Thus, our ability to rapidly identify synergistic and antagonistic drug interactions can potentially alter the patient outcomes.

*For correspondence:
jessica.brown@path.utah.edu

**Competing interests:** The authors declare that no competing interests exist.

## Introduction

Invasive fungal infections are an increasing problem worldwide, contributing to 1.6 million deaths annually (*Almeida et al., 2019*; *Bongomin et al., 2017*; *Brown et al., 2012*). These problematic infections are difficult to treat for many reasons. Delayed diagnoses, the paucity and toxicity of antifungal drugs, and the already immunocompromised state of many patients result in mortality rates of up to 90% (*Brown et al., 2012*; *Pianalto and Alspaugh, 2016*; *Scorzoni et al., 2017*). To date, there are only four classes of antifungals, which primarily target the fungal cell envelope (cell wall and plasma membrane) (*Coelho and Casadevall, 2016*; *Odds et al., 2003*; *Pianalto and Alspaugh, 2016*; *Scorzoni et al., 2017*). The population of immunocompromised individuals is growing due to medical advancements, such as immunosuppression for transplants and chemotherapy. Emerging fungal pathogens are simultaneously increasing in both clinical burden and the number of causal species due to human activity such as agricultural drug use (*Berger et al., 2017*) and global warming (*Almeida et al., 2019*; *Garcia-Solache and Casadevall, 2010*). Thus, the need for more and better antifungal therapeutics is evident.

Cryptococcosis is among the most common invasive mycoses, causing 220,000 life-threatening infections and 180,000 deaths annually worldwide (*Rajasingham et al., 2017*). *Cryptococcus neoformans* and *Cryptococcus gattii* are the etiological agents of cryptococcosis, though nearly 95% of cases are caused by *C. neoformans* (*Brown et al., 2012*; *Maziarz and Perfect, 2016*). As *C. neoformans* is globally distributed throughout the environment, most individuals are exposed by two years of age (*Goldman et al., 2001*). However, systemic disease primarily occurs in the immunocompromised, particularly those with decreased T helper-1 cell reactions (*Maziarz and Perfect, 2016*).

**eLife digest** Individuals with weakened immune systems – such as recipients of organ transplants – can fall prey to illnesses caused by fungi that are harmless to most people. These infections are difficult to manage because few treatments exist to fight fungi, and many have severe side effects. Antifungal drugs usually slow the growth of fungi cells rather than kill them, which means that patients must remain under treatment for a long time, or even for life.

One way to boost efficiency and combat resistant infections is to combine antifungal treatments with drugs that work in complementary ways: the drugs strengthen each other's actions, and together they can potentially kill the fungus rather than slow its progression. However, not all drug combinations are helpful. In fact, certain drugs may interact in ways that make treatment less effective. This is particularly concerning because people with weakened immune systems often take many types of medications.

Here, Wambaugh et al. harnessed a new high-throughput system to screen how 2,000 drugs (many of which already approved to treat other conditions) affected the efficiency of a common antifungal called fluconazole. This highlighted 19 drugs that made fluconazole less effective, some being antibiotics routinely used to treat patients with weakened immune systems.

On the other hand, 40 drugs boosted the efficiency of fluconazole, including dicyclomine, a compound currently used to treat inflammatory bowel syndrome. In fact, pairing dicyclomine and fluconazole more than doubled the survival rate of mice with severe fungal infections. The combined treatment could target many species of harmful fungi, even those that had become resistant to fluconazole alone.

The results by Wambaugh et al. point towards better treatments for individuals with serious fungal infections. Drugs already in circulation for other conditions could be used to boost the efficiency of fluconazole, while antibiotics that do not decrease the efficiency of this medication should be selected to treat at-risk patients.

Accordingly, HIV/AIDS patients account for 80% of cryptococcal cases (*Maziarz and Perfect, 2016*; *Rajasingham et al., 2017*).

The primary treatment for cryptococcosis involves three different classes of antifungals. Standard care is a combination of amphotericin B (polyene class) and 5-fluorocytosine (5-FC; pyrimidine analog) for two weeks, followed by high dose azole treatment (e.g. fluconazole (FLZ)) for at least 8 weeks, and finally a low dose oral FLZ for at least 6 months (*Cox and Perfect, 2018*; *Mourad and Perfect, 2018*). Despite this, mortality rates remain as high as 80% for cryptococcal meningitis (*Rajasingham et al., 2017*). This is mainly due to the difficulty of obtaining ideal treatment standards. 5-FC is unavailable in 78% of countries, mostly due to licensing issues (*Kneale et al., 2016*; *Mourad and Perfect, 2018*). Without the inclusion of 5-FC in the treatment regiment, mortality increases by up to 25% (*Kneale et al., 2016*). Amphotericin B is administered intravenously, requiring hospitalization. Treatment with amphotericin B is therefore particularly challenging in areas such as sub-Saharan Africa, which has the highest burden of cryptococcal disease (*Rajasingham et al., 2017*). Due to these therapeutic hurdles, many patients are treated with FLZ alone, which decreases survival rates from 75% to 30% in high burden areas (*Kneale et al., 2016*). Additional treatment options are thus needed to prevent these unnecessary deaths.

One theoretical approach to improve treatment is synergistic combination therapy. Synergistic interactions occur when the combined effect of two drugs is greater than the sum of each drug's individual activity (*Cokol et al., 2011*; *Kalan and Wright, 2011*). This is a powerful treatment option which has been utilized for a variety of infections (*Kalan and Wright, 2011*; *Robbins et al., 2015*; *Spitzer et al., 2011*; *Zheng et al., 2018*). For instance, the common combination of sulfamethoxazole and trimethoprim for bacterial infections is a synergistic interaction that works to inhibit sequential steps in bacterial folic acid biosynthetic pathway (*Kalan and Wright, 2011*). In addition, Amphotericin B and 5-FC act synergistically, and mortality rates increase dramatically when one is unavailable (*Beggs, 1986*; *Kneale et al., 2016*; *Schwarz et al., 2006*). Synergistic interactions can also cause fungistatic drugs to switch to fungicidal, providing a more effective treatment option (*Cowen et al., 2009*).

Additionally, molecules can interact antagonistically to decrease therapeutic efficacy (*Caesar and Cech, 2019*; *Roberts and Gibbs, 2018*). Bacterial growth increases when antibiotics that inhibit DNA synthesis are used in combination with protein synthesis inhibitors (*Bollenbach et al., 2009*). Such antagonistic interactions further complicate already challenging infections (*Khandeparkar and Rataboli, 2017*; *Vadlapatla et al., 2014*), since immunocompromised patients are frequently treated with multiple drugs. 56% of AIDS patients experience polypharmacy, or greater than five medications (*Siefried et al., 2018*). Polypharmacy doubles the risk of antiviral therapy nonadherence to 49% of HIV⁺ patients (*Lohman et al., 2018*) and increases mortality by 68% in HIV⁺ and 99% in HIV⁻ patients (*Cantudo-Cuenca et al., 2014*). Better understanding of the molecular mechanisms underlying both synergistic and antagonistic drug interactions will allow us to improve identification and selection for or against these interactions.

The overlap² method (O2M) uses at least one known synergistic drug pair and a large-scale chemical-genetics dataset to predict synergistic and antagonistic drug interactions rapidly and on large scales (*Brown et al., 2014*; *Wambaugh and Brown, 2018*; *Wambaugh et al., 2017*). Each molecule of the synergistic pair induces a growth phenotype in a precise set of mutants (enhanced or reduced growth). Since these mutants exhibit the same phenotype in the presence of both molecules in a synergistic pair, we hypothesize that any other molecule eliciting the same phenotype in those mutants will also synergize with each molecule in the original pair. This method can be used against multiple microbes and applied to any published chemical-genetics dataset (*Brown et al., 2014*; *Wambaugh et al., 2017*).

In this study, we utilized previously identified synergy prediction mutants (*Brown et al., 2014*) to screen a library of small molecules enriched for Federal Drug Administration (FDA-)approved molecules. We non-discriminately identified 59 molecules that interact with FLZ, either synergistically or antagonistically. When validating these new combinations, we found that even though the analysis used a *C. neoformans* dataset (*Brown et al., 2014*), our synergistic and antagonistic combinations acted against pathogenic fungi from multiple phyla. These include *C. deuterogattii*, *Candida* species, and multiple clinical and environmental strains of *C. neoformans*, as well as clinical isolates of the increasingly problematic and multi-drug resistant species *Candida auris* (*Chowdhary et al., 2017*). Furthermore, we elucidated molecular mechanisms underlying the interaction with FLZ for a few of our most clinically relevant combinations. We also demonstrate these effects in an in vivo model of cryptococcosis. A particularly promising synergistic combination, dicyclomine hydrochloride and FLZ, almost doubled time-to-endpoint in a murine infection model. In sum, our high-throughput method, O2M, identifies FLZ interacting molecules with potential clinical impacts.

## Results

### Synergy prediction mutants for fluconazole allow for high-throughput screening of small molecule interactions

We previously demonstrated that O2M identifies genes whose knockout mutants, termed synergy prediction mutants, exhibit phenotypes that are indicative of synergistic interactions between small molecules (*Brown et al., 2014*; *Wambaugh et al., 2017*). O2M identifies synergy prediction mutants by using a chemical-genetics dataset, in which a library of knockout mutants is grown in the presence of >100 small molecules. We calculated quantitative growth scores (slower or faster growth) for each mutant/molecule combination. This large number of phenotypes produces a 'chemical genetic signature' for each molecule in the dataset. We then compared the 'signatures' for fluconazole and known fluconazole synergizers to identify knockout mutants that exhibit significant phenotypes (i.e. statistically significant slow or fast growth) in all chemical genetic signatures (*Brown et al., 2014*). The rationale was that similarities between chemical-genetic signatures of known synergistic pairs contain information that is indicative of the synergistic interaction with fluconazole. We term these 'synergy prediction mutants' because their knockout phenotype predicts additional fluconazole-synergizing molecules. We then identified genes whose mutant exhibited significant growth scores ($|Z| > 2.5$) when compared to wild-type growth for fluconazole and each of its known synergizers. We compared the significant gene set for fluconazole and each synergizer and identified common genes. We called these 'synergy prediction mutants' (*Figure 1A*). Since fluconazole and its known synergizers induce a growth phenotype from these mutants, we hypothesized that any molecule eliciting

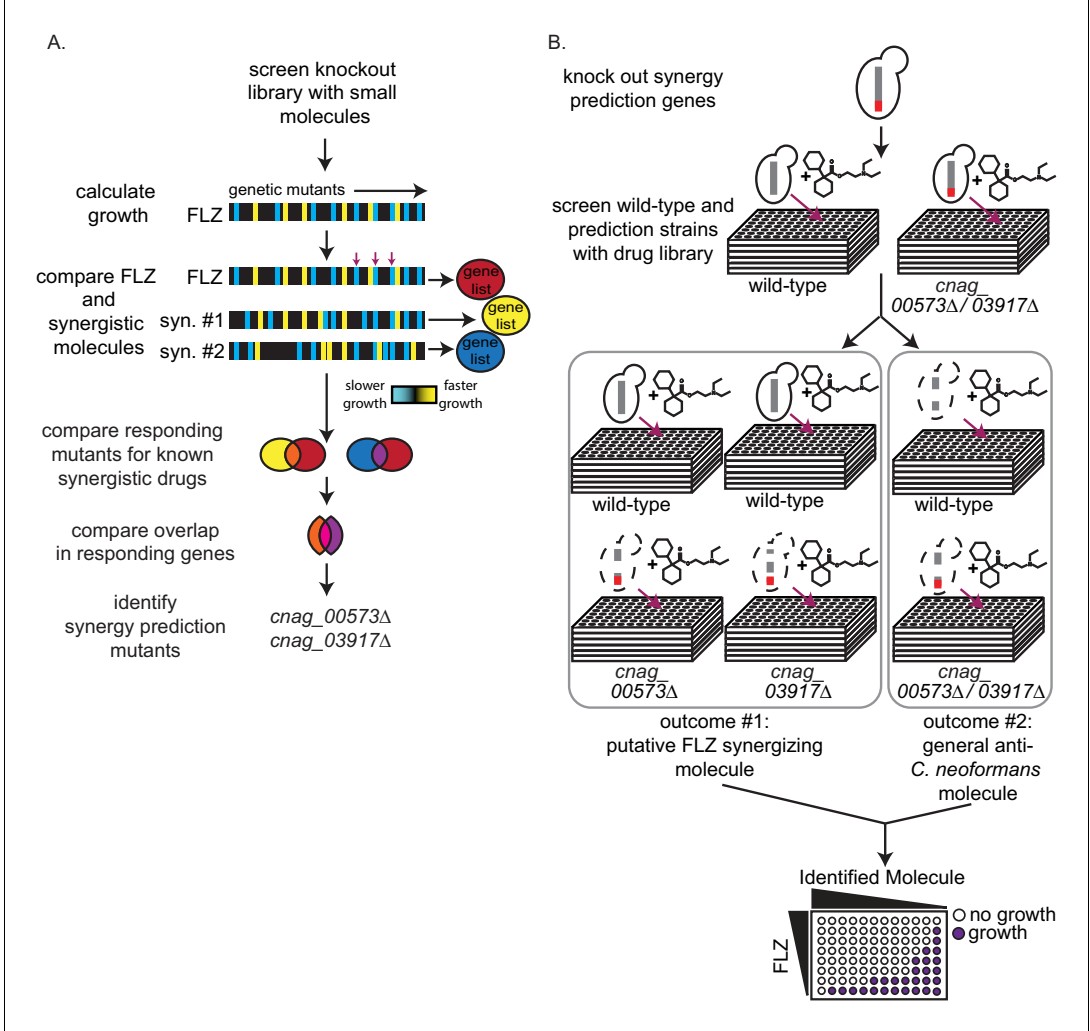

**Figure 1.** High-throughput screening for fluconazole interacting molecules using synergy prediction mutants. (**A**) Outline of overlap² method (O2M), which is also presented in Brown et al., Wambaugh et al., and Wambaugh and Brown. O2M requires a chemical-genetic dataset which can be generated by growing a collection of mutants in the presence of >100 small molecules individually. Growth scores are then calculated for each small molecule + mutant combination. In the heatmaps, the vertical line represents a different mutant. Blue represents slower growth compared to wild-type cells and yellow represents faster growth compared to wild-type cells. Comparing our starting drug (FLZ) and known synergistic molecules, we can identify genes whose knockout mutants show similar growth scores to the starting drug and all known synergistic partners (red arrows). These are the synergy prediction mutants. (**B**) Screening method to identify molecules that synergize with FLZ as well as anti-*C. neoformans* molecules. Synergy prediction mutants are created (red chromosome indicates gene knockout) and both synergy prediction mutant and wild type are grown in the library of small molecules (top section). Growth of wild type and synergy prediction mutants are accessed. Greyed out dotted yeast cell indicates differential growth compared to wild type (middle section). These molecules are then validated in a checkerboard assay (bottom section).

the same growth phenotype would also act synergistically with fluconazole. This was completed and tested in our previous publication (*Brown et al., 2014*) using FLZ and its known synergistic interacting partners fenpropimorph and sertraline (*Jansen et al., 2009*; *Zhai et al., 2012*). Our work identified three synergy prediction mutants (*cnag_00573Δ*, *cnag_03664Δ*, and *cnag_03917Δ*) (*Brown et al., 2014*). Since each chemical-genetic signature contains phenotypes from ~1400 gene knockouts, there does not have to be considerable overlap between chemical-genetic signatures to identify synergy prediction mutants.

Here we use synergy prediction mutants to rapidly screen for small molecules that synergize with fluconazole and can be quickly moved into clinical use. *CNAG 00573* encodes a NADH dehydrogenase (*Janbon et al., 2014*), *CNAG 03664* encodes NIC1, a high-affinity nickel-transporter (*Singh et al., 2013*), and *CNAG 03917* encodes a nuclear pore complex protein homologous to

Nup75 (*Liu et al., 2008*; *Stajich et al., 2012*). Using these gene mutants, we performed a high-throughput screen for synergistic interactions. Our assay is simple: differential growth between wild-type and synergy prediction mutants is indicative of a synergistic interaction with FLZ or any other starting drug. It does not require multi-drug assays, as the 'synergy prediction mutant' substitutes for one of the small molecules in the interaction, phenocopying the FLZ-small molecule interaction to produce synthetic lethality. We screened the Microsource Spectrum Collection, a small-molecule library of 2000 compounds enriched for FDA-approved molecules. We grew *C. neoformans* wild-type and synergy prediction mutants (*cnag_00573Δ* and *cnag_03917Δ*) in the presence of each small molecule (1 μM), identifying those that caused a significant difference in growth between the wild-type and both synergy prediction mutants after 48 hr of growth (*Figure 1B*). The mutant *cnag_03664Δ* was not used due to its inherent slow growth. The small molecule concentration of 1 μM gave the lowest false discovery rate when testing molecules known to synergize or not synergize with fluconazole. Using these synergy prediction mutants, we identified 313 putative FLZ synergistic molecules (*Supplementary file 1*).

We validated potential synergistic interactions in checkerboard assays, for which serial dilutions of each drug are crossed in a 96-well plate (*Figure 1B*). Synergistic interactions are defined as a $\geq 4$ fold decrease in the minimum inhibitory concentration (MIC) of each small molecule in the pair, resulting in a fractional inhibitory concentration index (FICI) of $\leq 0.5$ (*Johnson et al., 2004*; *Odds, 2003*). We tested the 129 molecules with single agent efficacy against *C. neoformans* growth in the preferred checkerboard assay. We found that 40 molecules were synergistic with FLZ, meaning 31% of these molecules were correctly predicted by O2M (*Figure 2A*). However, checkerboard assays require that both small molecules in the pair are able to inhibit growth of *C. neoformans* individually, which was not the case with all our putative synergistic molecules. In those cases, we performed Bliss Independence, which identifies whether molecules enhance the action of FLZ (*Tang et al., 2015*). In a 96-well plate, we created a gradient of FLZ combined with 10 μM and 100 nM concentrations of the 55 small molecules that could not inhibit *C. neoformans* alone. We found 6 of these molecules enhanced the action of FLZ at both 10 μM and 100 nM concentrations and were deemed synergistic (*Figure 2—figure supplement 1*). The FLZ-synergistic molecules belonged to a wide range of bioactive categories including antidepressants, adrenergic agonists, as well as antiinfectives (*Figure 2B* and *Table 1*).

Additionally, our screen identified antagonistic interactions. These interactions are defined by a minimum 4-fold increase in MICs causing increased fungal growth and a FICI $\geq 4$ (*Cetin et al., 2013*). We identified 19 antagonistic interactions with FLZ (*Figure 2C*). Of note, many antagonists were documented antiinfectives, including some antifungals (*Figure 2D* and *Table 1*). The remaining 70 molecules did not interact with FLZ and represent false positives of our screen (*Figure 2—figure supplement 2A*).

Overall, the O2M screen predicted that 16% of the library's small molecules would interact with FLZ. Of the predicted interactions, 46% were validated by checkerboard assay to truly interact with FLZ (synergistic or antagonistic) (*Figure 2E*). All small molecules that interact with FLZ and inhibit fungal cell growth (i.e. were tested in checkerboard assays) are listed with their MICs in *Table 1*. We determined a MIC for 90% growth inhibition (MIC$_{90}$) for most molecules. For a subset, we were only able to determine a MIC for 50% growth inhibition (MIC$_{50}$). In some cases, molecule solubility in aqueous solutions are too low to reach concentrations necessary to inhibit fungal growth. Alternatively, the inability to determine a MIC$_{90}$ could result from trailing growth, which has been seen in fungal species (*Arthington-Skaggs et al., 2002*). FLZ non-interacting molecules are listed in *Supplementary file 2*. The remaining 129 molecules predicted to interact were not tested due to unavailability, or known toxicities that would have made them impractical treatments.

## Identification of general anti-cryptococcal molecules by O2M

During the screening process, wild-type *C. neoformans* is grown in each of the small molecules alone, allowing us to identify general anti-*C. neoformans* molecules (*Figure 1B*). These molecules had MICs ranging from 16 nM to 760 μM, and were mostly listed as antifungals (*Supplementary file 3*). However, the phenotype of a general anti-*C. neoformans* molecule can overshadow any synergistic phenotypes in the synergy prediction mutant. Therefore, we also tested these molecules for synergistic interactions in the standard checkerboard assay. Two of the general anti-*C. neoformans*

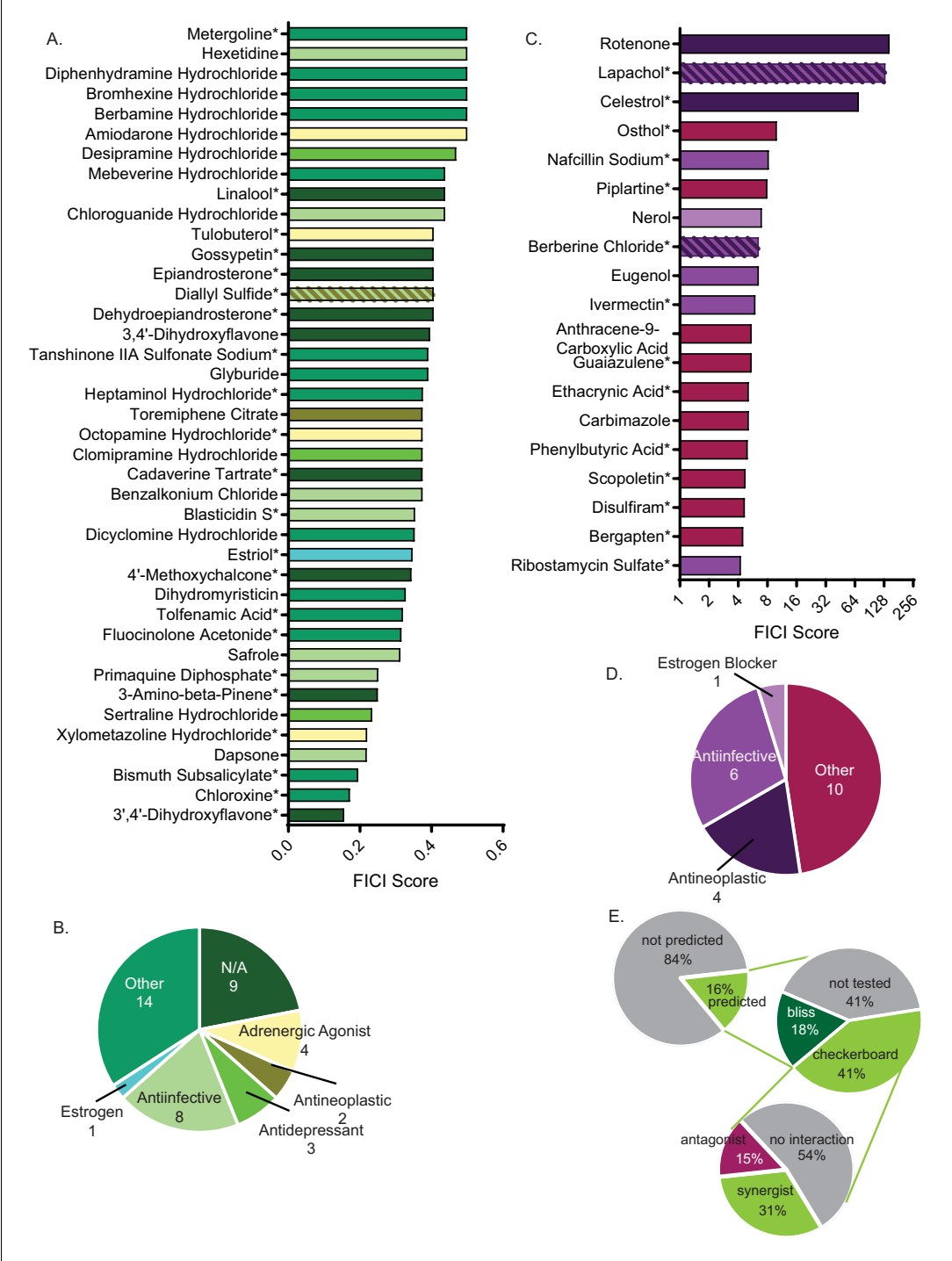

**Figure 2.** Synergistic and antagonistic molecules identified from high-throughput screen. (**A**) Average fractional inhibitory concentration index (FICI) score of synergistic molecules identified from our high-throughput screen. Color of bar corresponds with bioactivities listed in B. FICIs are listed in linear scale. Source data are in *Figure 2—source data 1*. (**B**) Categories of bioactivities of synergistic molecules with the corresponding number of molecules in each category. (**C**) FICI scores in log2 scale of antagonistic molecules from screen. Colors correspond with bioactivities listed in D. Source data are in *Figure 2—source data 2* (**D**) Categories of bioactivities of antagonistic molecules with corresponding number of molecules. All bioactivities came from Microsource Spectrum molecule list which is also seen in *Table 1*. (**E**) Representation of percentage of molecules from the entire Microsource Spectrum Library predicted to synergize with fluconazole based on screening results (top), molecules tested in various assays (middle), and molecules yielding an interaction from checkerboards (bottom). * represents FICI for 50% inhibition of *C. neoformans* (when 90% inhibition was not found). All other scores listed are the FICI for 90% inhibition (FICI90) unless otherwise stated. Molecules not tested were not available commercially. All

*Figure 2 continued on next page*

*Figure 2 continued*

average FICI scores represent an average of at least two independent tests, performed in our prior works (*Brown et al., 2014*; *Wambaugh and Brown, 2018*; *Wambaugh et al., 2017*). All data are against *C. neoformans* strain CM18.

The online version of this article includes the following source data and figure supplement(s) for figure 2:

**Source data 1.** FICI scores of synergistic small molecule combinations.
**Source data 2.** FICI scores of antagonistic small molecule combinations.
**Source data 3.** Bliss Independence scores of small molecule combinations.
**Source data 4.** FICI scores of non-synergistic small molecule combinations.
**Figure supplement 1.** Bliss independence scores of non-single agent molecules.
**Figure supplement 2.** FICI scores of non-interacting molecules and general antifungals.

molecules, sulconazole nitrate and tacrolimus, were synergistic with FLZ (*Figure 2—figure supplement 2B*).

## Molecular structure predicts additional synergistic interactions

Once we identified new FLZ-based synergistic pairs, we sought to predict additional FLZ-synergizing molecules in order to increase our chances of identifying a promising therapeutic combination. We hypothesized that structural similarity with our newly-identified FLZ-synergizing molecules would predict additional FLZ synergizers. We chose four new FLZ synergizers that contain large ring structures, which are common among our synergizers: dicyclomine HCl, desipramine HCl, sertraline HCl, and diphenhydramine HCl. We used both ChemSpider (*Pence and Williams, 2010*) and ChemMine MSC Similarity Tool to identify structurally similar molecules (*Backman et al., 2011*). We chose three molecules (proadifen, drofenine, and naftidrofuryl) with ≥50% structural similarity to dicyclomine HCl, and all were synergistic with FLZ in checkerboard assays (*Figure 3A–D,M*). We tested three molecules (impramine, mianserine, and lofepramine) with >40% structural similarity to desipramine, two of which synergized with FLZ (*Figure 3E–H,M*). Lofepramine, which was not synergistic, is the prodrug to desipramine. Sibutramine, shares >40% structural similarity with sertraline, and is synergistic with FLZ as well (*Figure 3I,J,M*). Lastly, we tested citalopram, which shares >45% structural similarity to diphenhydramine, but was not synergistic with FLZ (*Figure 3K–M*). Overall, 75% of the structurally similar molecules acted synergistically with FLZ, demonstrating that structure can serve as a powerful basis to predict additional synergistic combinations prior to elucidating mechanism of action.

## Fluconazole-synergizing and -antagonizing responses are conserved across fungal species

As our goal is to identify potential antifungal therapies with broad efficacy, we tested our new synergistic pairs against additional strains of *C. neoformans* and other medically important fungi. Since we identified numerous fluconazole interaction molecules (*Figure 2A*), we focused on several promising synergistic molecules based on either low MICs or interesting bioactivity. We tested these interactions against the *C. neoformans* lab strain KN99 and 10 environmental or clinical *C. neoformans* isolates (*Chen et al., 2015*). We also tested our combinations against *Cryptococcus deuterogattii*, *Candida albicans*, *Candida glabrata*, and two strains of *Candida auris* (*Supplementary file 4*). Our FLZ-synergizing small molecules displayed similar MICs across the different *C. neoformans* strains and fungal species (*Supplementary file 5*). Of the FLZ-interacting pairs, sertraline hydrochloride (HCl), clomipramine HCl, benzalkonium chloride, berbamine HCl and dicyclomine HCl synergistically inhibited the growth of most of the strains/species (*Figure 4A,B,C,D,E*). We also chose two antagonistic interactions to test in these additional strains and species based on bioactivities relevant to cryptococcosis patient populations. We chose nafcillin sodium, a common antibiotic, and Ivermectin, an antiparasitic drug (*Table 1* and *Figure 2C*). Both bacterial and parasitic infections are common among immunocompromised patients (*Kaplan et al., 2009*). Nafcillin sodium was antagonistic with FLZ in most of the strains/species (*Figure 4M*). The other synergistic or antagonistic molecule pairs had more variable results among the different strains/species (*Figure 4*), highlighting the importance of testing putative antimicrobials against multiple species and strains when considering therapeutic relevance.

**Table 1.** Minimum inhibitory concentrations for fluconazole interacting molecules.

All values are against *C. neoformans* strain CM18. MIC for 90% inhibition (MIC90) listed when possible. MIC50 = MIC for 50% inhibition. Molecules were dissolved in to their highest soluble concentration in DMSO. MICs were determined below the inhibitory concentrations of DMSO (~5%) in YNB minimal media.

| Small molecule | Bioactivity | MIC 50 (mM) | MIC 90 (mM) | Result |
|---|---|---|---|---|
| Anthracene-9-Carboxylic Acid | Cl transport inhibitor | 0.2 | 1.7 | antagonistic |
| Berberine Chloride | antiarrhythmic, alpha2 agonist, cholinesterase, anticonvulsant, antiinflammatory, antibacterial, antifungal, antitrypanosomal, antineoplastic, immunostimulant | 1.4 | 2.6 | antagonistic |
| Bergapten | antipsoriatic, antiinflammatory | 1.2 | 2.4 | antagonistic |
| Carbimazole | antithyroid | 0.2 | 0.9 | antagonistic |
| Celastrol | antineoplastic, antiinflamatory, NO synthesis inhibitor, chaperone stimulant | 0.07 | 0.2 | antagonistic |
| Disulfram | alcohol antagonist | 0.007 | 0.007 | antagonistic |
| Ethacrynic Acid | diuretic | 0.4 | 1.1 | antagonistic |
| Eugenol | analgesic (topical), antiseptic, antifungal | 2.4 | 14 | antagonistic |
| Guaiazulene | antioxidant, inhibits lipid peroxidation inhibitor, antiinflammatory, hepatoprotectant; LD50(rat) 1550 mg/kg po | 3.5 | 11 | antagonistic |
| Ivermectin | antiparasitic | 1.9 | 1.9 | antagonistic |
| Lapachol | antineoplastic, antifungal | 36 | N/A | antagonistic |
| Nafcillin Sodium | antibacterial | 2.4 | N/A | antagonistic |
| Nerol | weak estrogen receptor blocker | 0.2 | 0.7 | antagonistic |
| Osthol | N/A | 0.5 | 1.8 | antagonistic |
| Phenylbutyric Acid | antiinflammatory, antihyperammonemic (Na salt) | 0.7 | 2.9 | antagonistic |
| Piplartine | anti-asthma, antibronchitis | 3.3 | 190 | antagonistic |
| Ribostamycin Sulfate | antibacterial | 9.4 | 19 | antagonistic |
| Rotenone | acaricide, ectoparasiticide, antineoplastic, mitochondrial poison | 0.002 | 0.002 | antagonistic |
| Scopoletin | NO synthesis (inducible) inhibitor, anticoagulant | 2.9 | 11 | antagonistic |
| 3,4'-Dihydroxyflavone | N/A | 1.4 | 8.3 | synergistic |
| 3',4'-Dihydroxyflavone | N/A | 8.9 | N/A | synergistic |
| 3-Amino-beta-Pinene | N/A | 12 | 50 | synergistic |
| 4'-Methoxychalcone | N/A | 0.1 | N/A | synergistic |
| Amiodarone Hydrochloride | adrenergic agonist, coronary vasodilator, Ca channel blocker | 0.003 | 0.007 | synergistic |
| Benzalkonium Chloride | antiinfective (topical) | 0.015 | 0.015 | synergistic |
| Berbamine Hydrochloride | antihypertensive, skeletal muscle relaxant | 0.03 | 0.06 | synergistic |
| Bismuth Subsalicylate | antidiarrheal, antacid, antiulcer | 1.5 | N/A | synergistic |
| Blasticidin S | antibiotic, antifungal; LD50 (rat po) 16 mg/kg | 0.23 | 0.72 | synergistic |
| Bromhexine Hydrochloride | expectorant | 15 | 15 | synergistic |
| Cadaverine Tartrate | N/A | 2.3 | 5.8 | synergistic |
| Chloroguanide Hydrochloride | antimalarial | 1 | 1 | synergistic |
| Chloroxine | chelating agent | 0.002 | 0.007 | synergistic |
| Clomipramine Hydrochloride | antidepressant | 0.65 | 0.78 | synergistic |
| Dapsone | antibacterial, leprostatic, dermatitis herpetiformis suppressant | 29 | N/A | synergistic |
| Dehydroepiandrosterone | N/A | 1.5 | 12 | synergistic |
| Desipramine Hydrochloride | antidepressant | 1.9 | 1.9 | synergistic |
| Diallyl Sulfide | antibacterial, antifungal, antineoplastic, antihypercholesterolaemic, hepatoprotectant | 7.2 | 33 | synergistic |
| Dicyclomine Hydrochloride | anticholinergic | 4.9 | 4.9 | synergistic |

*Table 1 continued on next page*

*Table 1 continued*

| Small molecule | Bioactivity | MIC 50 (mM) | MIC 90 (mM) | Result |
|---|---|---|---|---|
| Dihydromyristicin | GSH transferase inducer | 280 | N/A | synergistic |
| Diphenhydramine Hydrochloride | antihistaminic | 18 | N/A | synergistic |
| Epiandrosterone | N/A | 1.2 | 3.2 | synergistic |
| Estriol | estrogen | 25 | 25 | synergistic |
| Fluocinolone Acetonide | glucocorticoid, antiinflammatory | 1.8 | 3.7 | synergistic |
| Glyburide | antihyperglycemic | 0.3 | 2 | synergistic |
| Gossypetin | N/A | 1.6 | N/A | synergistic |
| Heptaminol Hydrochloride | vasodilator | 6.2 | N/A | synergistic |
| Hexetidine | antifungal | 0.05 | 0.3 | synergistic |
| Linalool (+) | N/A | 0.8 | 3.1 | synergistic |
| Mebeverine Hydrochloride | muscle relaxant (smooth) | 4.4 | 8.9 | synergistic |
| Metergoline | analgesic, antipyretic | 0.2 | 0.2 | synergistic |
| Octopamine Hydrochloride | adrenergic agonist | 25 | N/A | synergistic |
| Primaquine Diphosphate | antimalarial | 3.7 | N/A | synergistic |
| Safrole | anesthetic (topical) and antiseptic, pediculicide | 20 | 41 | synergistic |
| Sertraline Hydrochloride | antidepressant, 5HT uptake inhibitor | 0.2 | 0.3 | synergistic |
| Tanshinone IIA Sulfonate Sodium | free radical scavenger | 0.9 | 1.2 | synergistic |
| Tolfenamic Acid | antiinflammatory, analgesia | 0.8 | 7.7 | synergistic |
| Toremiphene Citrate | antineoplastic, anti-estrogen | 28 | 28 | synergistic |
| Tulobuterol | bronchodilator, beta adrenergic agonist | 23 | N/A | synergistic |
| Xylomethazoline Hydrochloride | adrenergic agonist, nasal decongestant | 2.4 | N/A | synergistic |

## Exposure to beta-lactam antibiotics increases ergosterol levels and antagonizes fluconazole activity

Upon identifying FLZ interacting combinations that are consistent among other fungal species and strains, we sought to investigate a potentially harmful combination for treating fungal infections. Among the antagonists, we were particularly interested in the antagonistic interaction between nafcillin sodium (nafcillin) and FLZ that emerged in multiple fungal strains and species (*Figure 4M*). Nafcillin is a common penicillinase-resistant penicillin antibiotic (*Letourneau, 2019*; *Nathwani and Wood, 1993*). Furthermore, the cryptococcosis patient population, which consists of mainly HIV/AIDS patients, is at high risk for multiple infections, increasing the likelihood that they could require overlapping treatments for bacterial and fungal infections (*Kaplan et al., 2009*). We first tested the nafcillin + FLZ combination on additional clinical isolates of either *C. neoformans* or *Candida auris* that are considered FLZ resistant (*Supplementary files 4* and *6*). The FLZ MIC for resistant *C. neoformans* strains ranged from 1 to 32 μg/mL and was 256 μg/mL for *C. auris* upon initial resistance characterization. In all 7 *C. neoformans* strains, nafcillin antagonized FLZ activity. Nafcillin acted antagonistically with FLZ against one of the three *C. auris* strains (*Figure 4—figure supplement 1* and *Figure 4—source data 5*).

We next asked whether other beta-lactam antibiotics also antagonize FLZ activity (*Figure 5A–L*). Antagonism with FLZ was not a universal attribute among penicillin antibiotics but was evident for oxacillin and methicillin (*Figure 5M*). We also tested a first-generation cephalosporin, cefazolin, that is often prescribed in place of nafcillin (*Letourneau, 2019*; *Miller et al., 2018*) and a second-generation cephalosporin, cefonicid. We found that both of these molecules also antagonize FLZ (*Figure 5M*). Of the beta-lactams tested, those that are often used to treat *Staphylococcus aureus*

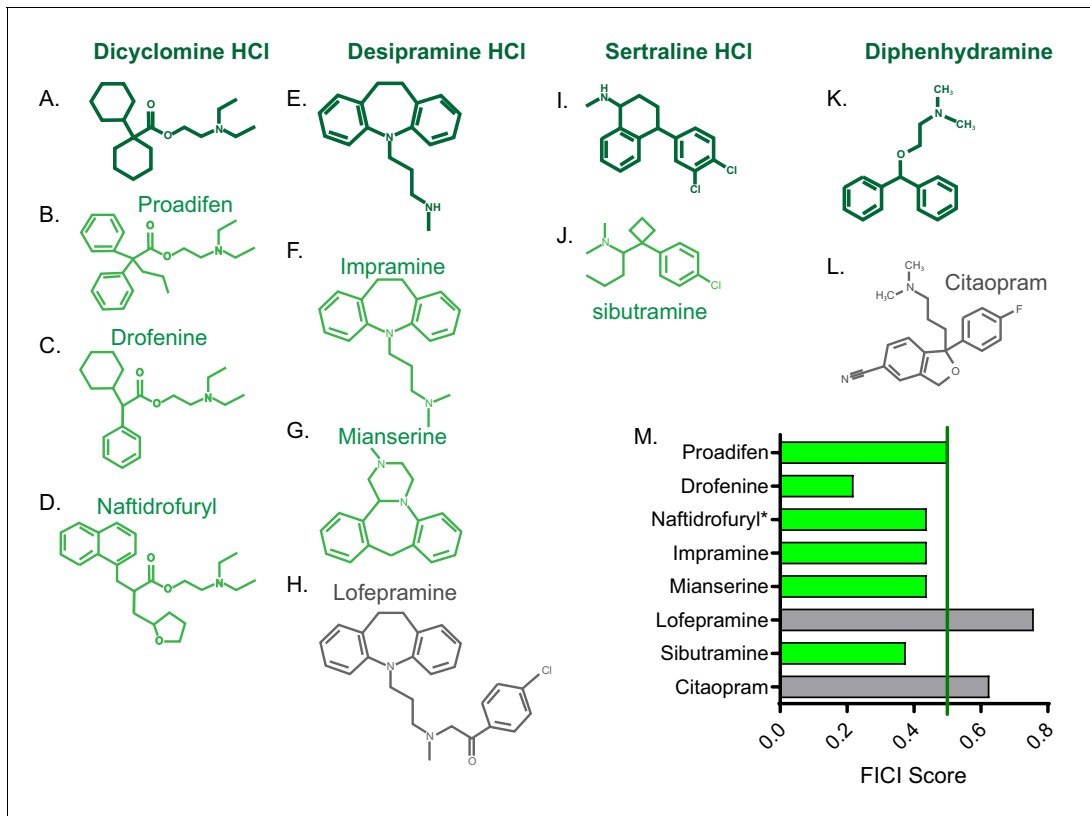

**Figure 3.** FICI scores of structurally similar molecules. Chemical structure of (**A**) dicyclomine HCl and structurally similar molecules (**B**) proadifen, (**C**) drofenine, and (**D**) naftidrofuryl. Chemical structure of (**E**) desipramine HCl and structurally similar molecules (**F**) impramine, (**G**) mianserine, and H) lofepramine. Chemical structure of (**I**) sertraline HCl and structurally similar (**J**) sibutramine. Chemical structure of (**K**) diphenhydramine HCl and structurally similar (**L**) citalopram. (**M**) Fractional inhibitory concentration index (FICI) score of structurally similar molecules. Synergistic interactions with FLZ labeled in green and non-interacting molecules labeled in grey. * represents FICI for 50% inhibition of *C. neoformans* strain CM18 all others listed are FICI for 90% inhibition. Average FICI scores represent a minimum of two independent replicates. All average FICI scores represent an average of at least two independent tests. Source data are *Figure 3—source data 1*.

The online version of this article includes the following source data for figure 3:

**Source data 1.** FICI scores of small molecules with structures similar to newly identified fluconazole-synergizers.

were antagonistic with FLZ (*Fowler and Holland, 2018*; *Letourneau, 2019*). We therefore decided to test additional antibiotics used to treat *S. aureus* infections. These included vancomycin and linezolid (*Fowler and Holland, 2018*; *Lowy, 2019*), neither of which interacted with FLZ (*Figure 5M*).

To investigate the mechanism of antagonism, we looked to other known drug interactions involving nafcillin. In particular, nafcillin antagonizes warfarin and other drugs in vivo by inducing cytochrome P450 enzymes through an unknown mechanism, which increases warfarin metabolism (*King et al., 2018*; *Wungwattana and Savic, 2017*). FLZ inhibits a fungal cytochrome P450 enzyme, 14α-demethylase, which halts ergosterol biosynthesis and fungal growth (*Figure 5N*; *Odds et al., 2003*). We hypothesized that nafcillin may also induce cytochrome P450 enzymes, such as 14α-demethylase, in *C. neoformans*, counteracting FLZ's mechanism of action. To test this hypothesis, we examined whether nafcillin affects ergosterol biosynthesis. We extracted sterols from *C. neoformans* cells grown in the presence of nafcillin, FLZ, nafcillin + FLZ, or vehicle. We found an increase in ergosterol with a high-dose nafcillin treatment (*Figure 5O*). Ergosterol levels in nafcillin + FLZ are not statistically different from the control treatment (*Figure 5O*). Next, we tested whether nafcillin is synergistic with the antifungal amphotericin B. Since amphotericin B kills target cells by binding and extracting ergosterol from the plasma membrane (*Anderson et al., 2014*), we hypothesized that if nafcillin increases ergosterol levels in *C. neoformans*, nafcillin would act synergistically with amphotericin B by increasing amphotericin B binding sites (i.e. ergosterol). Indeed, amphotericin B was synergistic with nafcillin in checkerboard assays giving an average FICI score of 0.5 (*Figure 5—source*

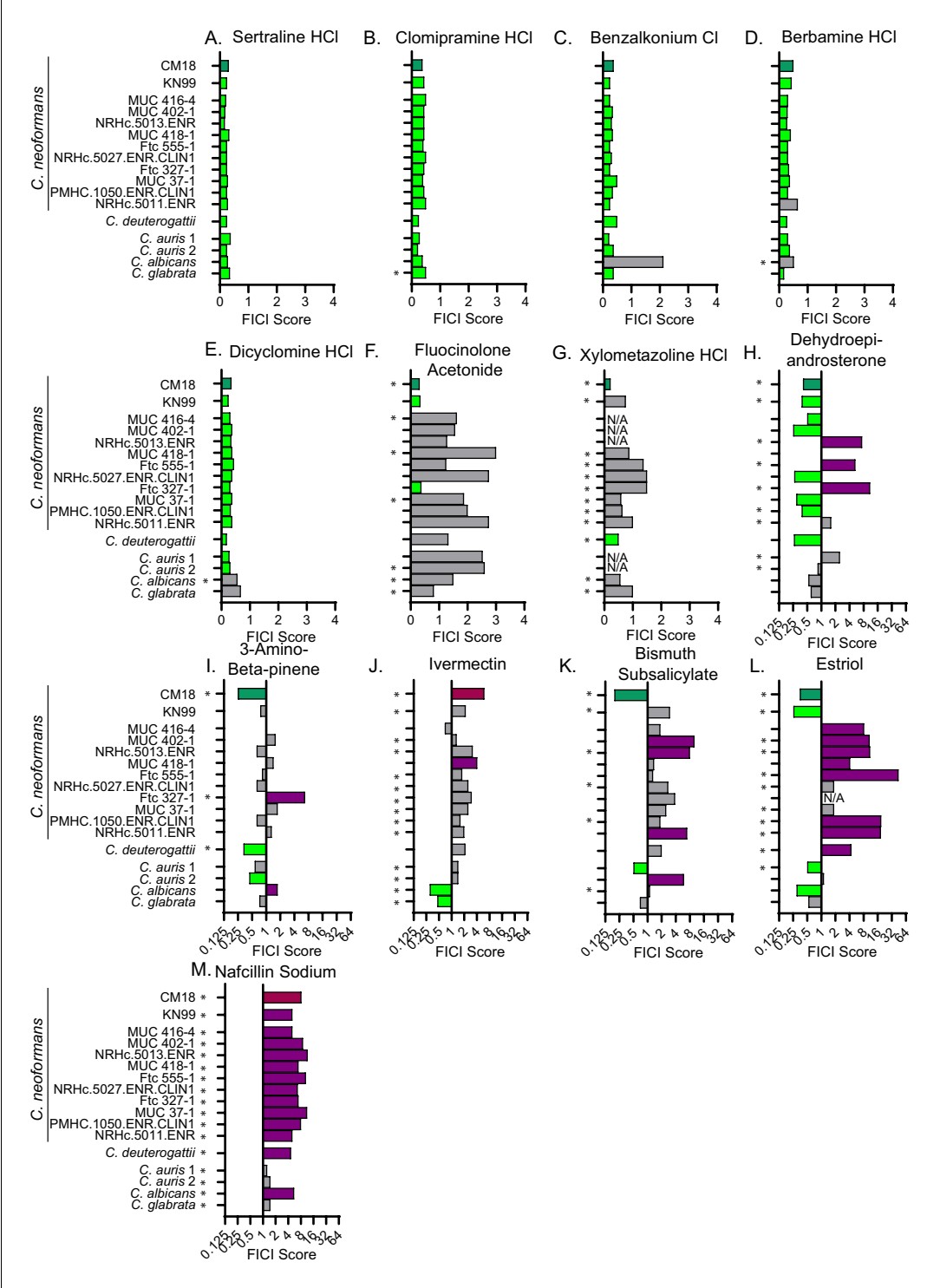

**Figure 4.** Synergistic and antagonistic combinations affect other fungal strains and species. Fractional inhibitory concentration index (FICI) scores of synergistic and antagonistic combinations with FLZ in other fungal strains/species for (**A**) Sertraline (**B**) Clomipramine HCl (**C**) Benzalkonium Cl (**D**) Berbamine HCl (**E**) Dicyclomine HCl (**F**) Fluocinolone Acetonide (**G**) Xylometazoline HCl (**H**) Dehydroepiandrosterone (**I**) 3-Amino-beta-pinene (**J**) Ivermectin (**K**) Bismuth Subsalicylate (**L**) Estriol (**M**) Nafcillin Sodium. * represents FICI for 50% inhibition all other scores listed are the FICI90. Strains/species listed on left and **Supplementary file 4**. CM18 (top) represents original result. Green bars represent FICI scores ≤ 0.5 yielding a synergistic result. Violet bars represent FICI scores ≥ 4 yielding an antagonistic result. No interactions are in grey bars. FICI Scores presented in either linear or log2 scale. HCl = hydrochloride, Cl = chloride. All average FICI scores represent an average of at least two independent tests. Source data are **Figure 4—source data 1**.

*Figure 4 continued on next page*

*Figure 4 continued*

The online version of this article includes the following source data and figure supplement(s) for figure 4:

**Source data 1.** FICI scores of synergistic small molecule combinations against a variety of fungal species and strains.
**Source data 2.** FICI scores of nafcillin in combination with ketoconazole.
**Source data 3.** FICI scores of dicyclomine in FLZ resistant strains and species.
**Source data 4.** FICI scores of dicyclomine in combination with ketoconazole.
**Source data 5.** FICI scores of nafcillin in FLZ resistant strains and species.
**Figure supplement 1.** Nafcillin is antagonistic in most FLZ resistant strains and species.
**Figure supplement 2.** Dicyclomine is synergistic in most FLZ resistant strains and species.

*data 2*), further suggesting that nafcillin increases ergosterol levels in *C. neoformans.* Lastly, ketoconazole is an imidazole antifungal that, like fluconazole, inhibits ergosterol synthesis (*Marichal et al., 1985*; *Odds et al., 2003*) and is also a potent inhibitor of antiretroviral drug metabolism through its inhibition of cytochrome P450 enzymes (*Dooley et al., 2008*; *Kaeser et al., 2009*; *Polk et al., 1999*). Ketoconazole is used in certain areas worldwide in order to increase the half-life of antivirals in blood (*Dooley et al., 2008*). We found that nafcillin also antagonizes ketoconazole activity, producing an average FICI score of 5 against *C. neoformans* (*Figure 4—source data 2*). Overall, nafcillin and other beta-lactam antibiotics could decrease treatment efficacy when combined with antifungals FLZ and ketoconazole.

## The synergistic interaction between dicyclomine HCl and FLZ affects cell permeability and nutrient uptake

Finally, we investigated a promising fluconazole-synergizer that was effective against multiple fungal species and strains: dicyclomine HCl (dicyclomine), an anticholinergic agent (*Table 1*). To further evaluate the potential of the combination, we first tested this molecule in the FLZ-resistant *C. neoformans* and *Candida auris* strains (*Supplementary files 4* and *6*). Dicyclomine was synergistic with FLZ against each of these resistant strains and species (*Figure 4—figure supplement 2*). We also tested whether dicyclomine synergizes with ketoconazole. As previously mentioned, ketoconazole has effects in combination with antiretroviral therapies and maybe used in certain areas of HIV endemicity. We found the dicyclomine + ketoconazole combination was also synergistic in a checkerboard assay, giving an average FICI score of 0.313 (*Figure 4—source data 4*).

We next investigated the molecular mechanism underlying the dicyclomine + FLZ interaction. Dicyclomine is an FDA-approved drug, also known as Bentyl, used to treat urinary incontinence (*Malone and Okano, 1999*; *Page and Dirnberger, 1981*). It targets a G-protein coupled receptor (GPCR) encoded by the *CHRM1* gene (*UniProt Consortium., 2018*; *Kilbinger and Stein, 1988*). *C. neoformans* does not have a *CHRM1* ortholog, but there are a large number of GPCRs in fungi that could be potential targets (*Xue et al., 2008*). When we screened a deletion library of *C. neoformans* for mutants resistant to dicyclomine, we found that 44% of the annotated dicyclomine-resistant mutants were involved in transport and trafficking, suggesting that those processes may be related to dicyclomine's mechanism (*Figure 6A* and *Supplementary file 7*). Thus, we hypothesized that dicyclomine alters Golgi transport. In *Saccharomyces cerevisiae*, simultaneously inhibiting Golgi trafficking and blocking ergosterol synthesis leads to mislocalization of essential plasma membrane transporters (*Estrada et al., 2015*). We hypothesized that the combination of dicyclomine and FLZ will phenocopy this effect in *C. neoformans* (*Figure 6B*). Using propidium iodide internalization as a measure of cell permeability, we observed that dicyclomine, similar to FLZ, permeabilizes fungal cells at high doses (*Figure 6C,D,F,G*). Furthermore, we saw a greater than additive increase in permeability when fungal cells were treated with low concentrations of dicyclomine and FLZ in combination (*Figure 6E and H*). Dicyclomine-induced permeability appeared to be independent of significant changes to cell wall chitin (*Figure 6—figure supplement 2A-F*).

We next tested if dicyclomine + FLZ disrupted nutrient transporter function by measuring uptake of amino acids. If amino acid permeases are not localized to the plasma membrane, fungal cells are resistant to toxic amino acid analogs (*Roberg et al., 1997*). Using the same low doses of FLZ and DIC that alone do not permeabilize cells, *C. neoformans* is susceptible to the effect of either 5-fluoroanthranilic acid (5-FAA) or 5-methyl-tryptophan (5-MT). When the dose is combined, cells now

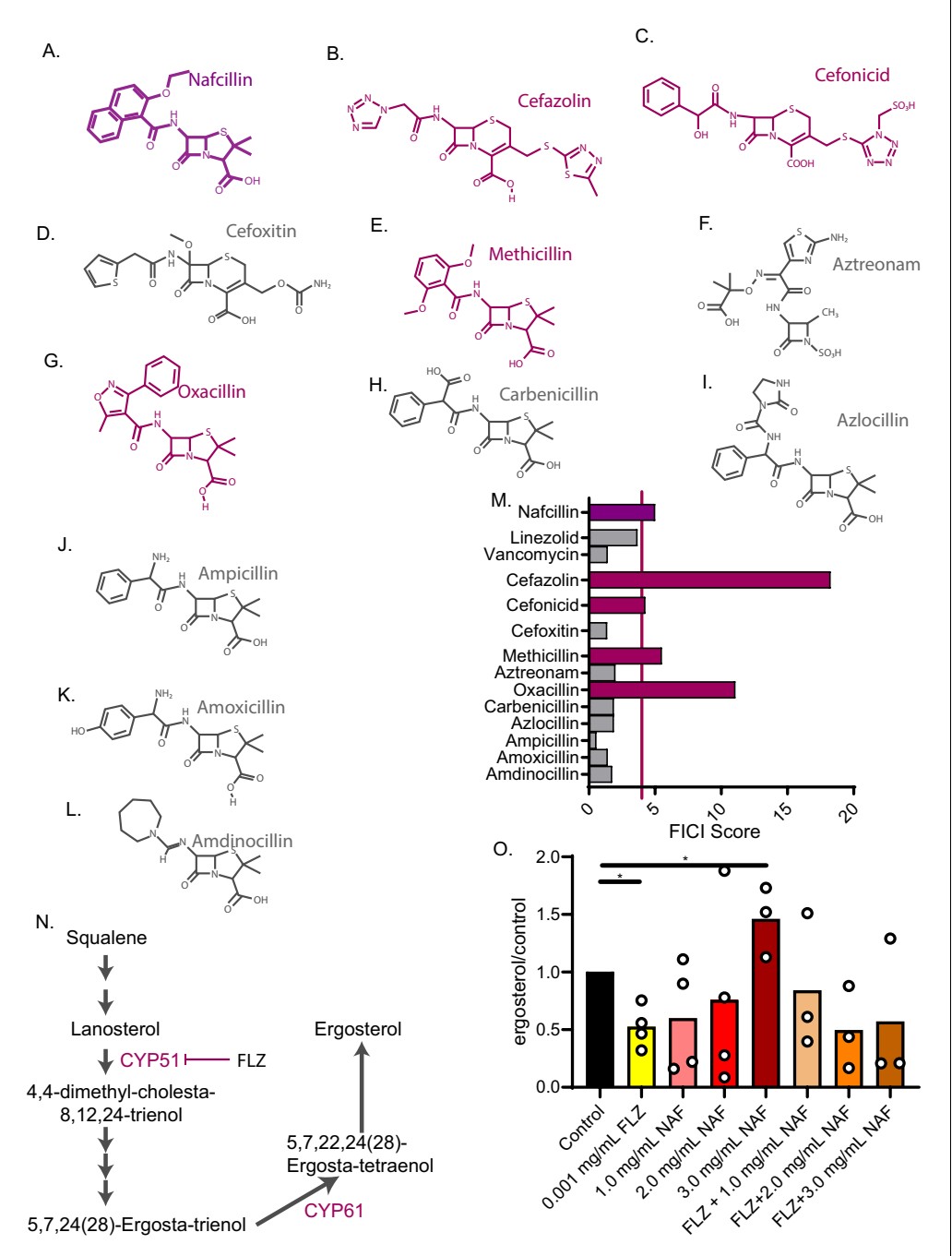

**Figure 5.** Nafcillin Sodium affects ergosterol levels. Molecular structures of beta-lactam antibiotics shown for (**A**) Nafcillin Sodium (**B**) Cefazolin (**C**) Cefonicid (**D**) Cefoxitin (**E**) Methicillin (**F**) Aztreonam (**G**) Oxacillin (**H**) Carbenicillin (**I**) Azlocillin (**J**) Ampicillin (**K**) Amoxicillin (**L**) Amdinocillin. (**M**) FICI scores for 50% inhibition of *C. neoformans* of various antibiotics related to nafcillin sodium tested with fluconazole. Violet bars over the red line illustrate a FICI score of ≥4 indicating antagonism. (**N**) Ergosterol biosynthesis pathway illustrating cytochrome P450 enzymes. (**O**) Ergosterol quantification from cell treated with Nafcillin (NAF), FLZ, or NAF+FLZ. Data normalized to control treated. *=p value is 0.03 (Mann-Whitney test). All average FICI scores represent an average of at least two independent tests (technical and biological replicates). Source data are in *Figure 5—source data 1*.

The online version of this article includes the following source data for figure 5:

**Source data 1.** FICI scores of small molecules with structures similar to nafcillin.

**Source data 2.** FICI scores of nafcillin in combination with amphotericin B.

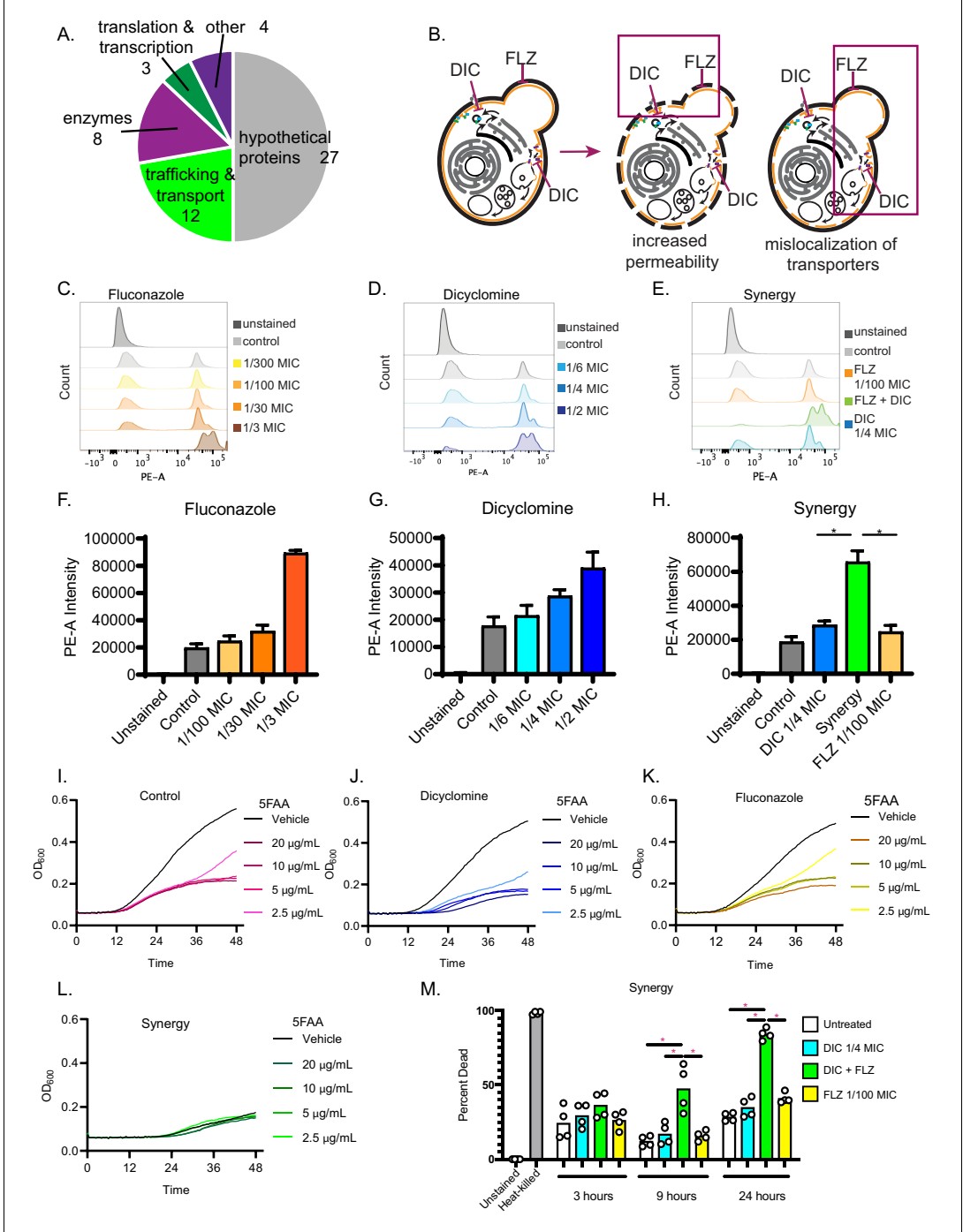

**Figure 6.** Dicyclomine affects permeability and nutrient transporters. (A) Pie chart with processes of deletion mutants that were resistant to dicyclomine. Numbers represent number of mutants. (B) Prediction for dicyclomine (DIC) + FLZ synergy mechanism. (C–E) Representative flow plots of propidium iodide staining. (F–H) Quantification of propidium iodide staining. Data are averages of three independent replicates. (I–L) Growth curves of *C. neoformans* with and without various concentrations of 5-FAA in addition to control (1x YNB + 2% glucose), dicyclomine (0.3 mg/mL or ¼ MIC), fluconazole (0.1 μg/mL or 1/100 MIC), or synergy (0.3 mg/mL dicyclomine + 0.1 μg/mL FLZ) treatment. Each experiment contained four technical replicates that were inoculated from the same culture. The lines represent the average of two experiments are presented in the figure. Source data are in *Figure 6—source data 1*. (M) Quantification of percent of dead cells after treatment with dicyclomine, FLZ, or synergy after 3, 9, and 24 hr. *=p value is 0.0286 (Mann-Whitney).

The online version of this article includes the following source data and figure supplement(s) for figure 6:

**Source data 1.** Growth rate of cells grown in the presence of toxic amino acid analog 5-FAA.

*Figure 6 continued on next page*

*Figure 6 continued*

**Source data 2.** CFUs after heat-killing *C. neoformans.*
**Figure supplement 1.** Dicyclomine additional effects on fungal chitin staining and nutrient intake.
**Figure supplement 1—source data 1.** Growth rate of cells grown in the presence of toxic amino acid analog 5-MT.
**Figure supplement 2.** Fungicidal effects of dicyclomine and combination treatment.

show resistance (*Figure 6I–L* and *Figure 6—figure supplement 1G–J*). This demonstrates that certain amino acid transporters' function is decreased and they thus may be mislocalized, conferring resistance to toxic forms of tryptophan.

Finally, we tested whether dicyclomine is able to kill *C. neoformans* cells, either alone or when combined with FLZ. FLZ is fungistatic, inhibiting the growth the *C. neoformans* cells but not killing them (*Hazen, 1998*; *Klepser et al., 1998*). Since synergistic interactions can cause fungistatic drugs to switch to fungicidal (*Cowen et al., 2009*), we tested whether our new synergistic combination of dicyclomine + FLZ killed *C. neoformans* cells. Using the stain BCECF-AM to identify dead cells (*McMullan et al., 2015*), we evaluated various doses of dicyclomine or FLZ individually or in combination. As a control, we tested the fungicidal drug amphotericin B (*Gray et al., 2012*). Dicyclomine only caused cell death after 3 hr at a very high dose (*Figure 6—figure supplement 2A,E*). We saw no increase in death of *C. neoformans* cells after treatments of FLZ after 3 hr (*Figure 6—figure supplement 2B,F*). However, when treating cells with 8 μg/mL amphotericin B nearly 100% of cells died within 3 hr (*Figure 6—figure supplement 2D,H*). After a 3 hr treatment, we did not observe an increase in cell death with the combination (*Figure 6—figure supplement 2C,G*). However, since dicyclomine + FLZ increases cell permeability and decreases nutrient uptake, we hypothesized that it may cause a slower death of cells than a fungicidal drug such as amphotericin B. Thus, we also sought to determine fungal cell death after 3, 9, and 24 hr of treatment. After 9 hr of treatment, we found that *C. neoformans* cells treated with all synergistic combinations and doses of FLZ 1/3 MIC and higher exhibited increased cell death (*Figure 6—figure supplement 2J*). After 24 hr of treatment, all synergistic combinations exhibited nearly 100% cell death. Additionally, high doses of FLZ also had killed almost all cells (*Figure 6—figure supplement 2K*). When evaluating the combination with the lowest FLZ + dicyclomine doses, we did not observe cell death at 3 hr. However, after 9- and 24 hr treatment, the combination had significantly more cell death than untreated or single agent-treated cells (*Figure 6—figure supplement 2M*; *Figure 6M*). Thus, our synergistic combination is fungicidal.

## Dicyclomine + FLZ act synergistically in vivo and enhances survival of mice with cryptococcosis

Since dicyclomine + FLZ combination exhibit a potent synergistic interaction in vitro, we tested its efficacy in a mouse model of cryptococcal meningitis. We intranasally inoculated outbred CD-1 mice (Charles River) with *C. neoformans* and allowed the infection to progress for 8 days. Colony forming unit data indicated that at this point 100% of the mice exhibited fungal dissemination to the liver, and 40% exhibited dissemination to the brain (*Figure 7—figure supplement 1*). A disseminated infection is consistent with human patients at treatment onset, as patients often don't seek treatment until *C. neoformans* has disseminated to the brain (*Zhu et al., 2010*). Between 8- and 40 days post-inoculation (d.p.i), we intraperitoneally administered dicyclomine, FLZ, dicyclomine + FLZ, or PBS (vehicle) daily. We used doses of both FLZ and dicyclomine which were within the range of doses given to humans (*Lexicomp, 2019a*; *Lexicomp, 2019b*). We sacrificed mice when they reached 80% of their initial mass (survival endpoint). Dicyclomine alone did not affect mouse survival compared to PBS-treatment. However, dicyclomine in combination with FLZ significantly improved survival over FLZ alone in a dose-dependent manner (*Figure 7*), indicating that dicyclomine is not effective at treating cryptococcosis on its own, but could well be of therapeutic benefit when combined with FLZ.

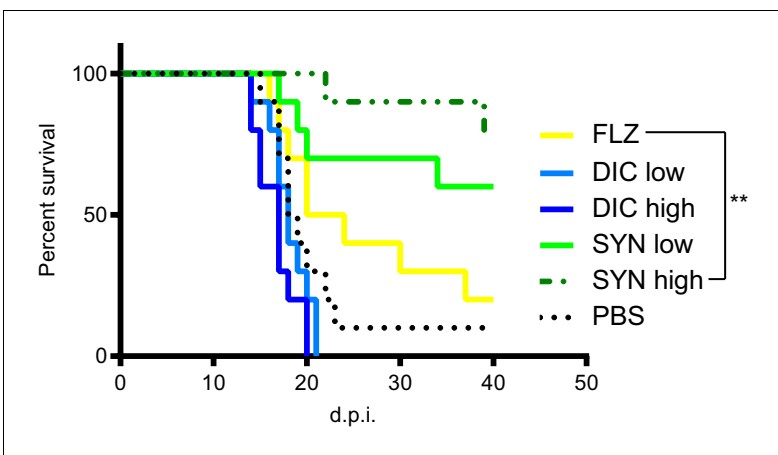

**Figure 7.** Dicyclomine + FLZ synergy increases survival of mice with cryptococcosis Survival of CD-1 outbred mice given FLZ (8 mg/kg), DIC low (1.15 mg/kg), DIC high (2.30 mg/kg), Synergy (SYN) low (FLZ + 1.15 mg/kg DIC), SYN high (FLZ + 2.30 mg/kg DIC) or PBS treatments.  N = 10. ; **=p value is 0.0036 (Mantel-Cox test).
The online version of this article includes the following figure supplement(s) for figure 7:

**Figure supplement 1.** *C. neoformans* disseminates by 8 days in CD-1 outbred mice.

## Discussion

Synergistic combination therapies are increasingly important clinical options, especially for drug resistant microbes (*Cowen and Lindquist, 2005*; *Kalan and Wright, 2011*; *Uppuluri et al., 2008*; *Zheng et al., 2018*). Traditionally, synergistic drug pairs were discovered serendipitously, but new methods are improving our ability to uncover important interactions (*Brown et al., 2014*; *Cokol et al., 2011*; *Cokol et al., 2018*; *Jansen et al., 2009*; *Robbins et al., 2015*; *Spitzer et al., 2011*; *Wambaugh et al., 2017*; *Wildenhain et al., 2015*). In this study, we identified a wide variety of molecules that interact synergistically with the antifungal FLZ to inhibit fungal growth. We do so without the use of noisy multi-drug assays, allowing for rapid and scalable screening. We also identify and investigate antagonistic interactions, which are clinically important (*Khandeparkar and Rataboli, 2017*; *Vadlapatla et al., 2014*) but have not been investigated in a systematic manner. This work and our application of O2M to antibiotic trimethoprim (*Wambaugh et al., 2017*) suggests that O2M's synergy prediction genes tend to identify synergizers with mechanism of action functionally related to those of the starting synergizer. Here, one of the known synergizers used to identify synergy prediction genes, sertraline, inhibits phospholipase activity in *Saccharomyces cerevisiae*, which inhibits vesicle formation *Rainey et al., 2010*). This could limit the range of synergizers identified by O2M. However, our studies on the trimethoprim demonstrates that identifying synergizers that phenocopy the downstream effect of known synergistic pairs, but target different factors, will bypass resistance to the starting synergizers due to the new targets (*Wambaugh et al., 2017*).

Of the 59 FLZ interacting molecules we identified, 10 have been previously described in various fungi. Of those 10, three were reported as synergists but antagonized FLZ activity in our assays (*Ahmad et al., 2010*; *Butts et al., 2014*; *Cardoso et al., 2016*; *Eldesouky et al., 2018*; *Kang et al., 2010*; *Li et al., 2018*; *Marchetti et al., 2000*; *Quan et al., 2006*; *Robbins et al., 2015*; *Spitzer et al., 2011*; *Zhai et al., 2012*). Prior work found that a single small molecule can both synergize with and antagonize the activity of a second small molecule depending on the concentration (*Meletiadis et al., 2007*). This phenomenon could explain the difference in some of our results compared to previous published interactions.

An important step in drug discovery is the ability to improve upon the efficacy of known drugs through synthetic modification and/or identification of structurally related molecules. We found that structural similarity predicts synergistic interactions (*Figure 3*), just as drugs of similar structure have similar function. These data demonstrate that many additional synergizers and antagonizers can be identified from a single example.

Furthermore, we identified broad spectrum interactions. All the combinations tested showed efficacy against multiple clinical and environmental isolates of *C. neoformans*, as well as *C. deuterogattii*, a related species which can cause disease in apparently immunocompetent individuals (*Applen Clancey et al., 2019*). We also tested our combinations against common *Candida* species that often develop multi-drug resistance (*Colombo et al., 2017*), including the emerging MDR pathogen *Candida auris* (*Figure 4*). Our data demonstrate that FLZ synergizers and antagonizers exhibit broad activities against multiple species and isolates.

We investigated the antagonistic interaction between FLZ and nafcillin, a beta-lactam antibiotic commonly used against *Staphylococcus aureus* and other difficult-to-treat bacterial infections (*Letourneau, 2019*). Patients with these infections include some of the same patients at risk for cryptococcosis (HIV and cancer patients) (*Kaplan et al., 2009*; *Utay et al., 2016*) and other fungal infections, including *C. auris* (*Rudramurthy et al., 2017*). Nafcillin can reach mean plasma concentrations in humans of 30 μg/mL after a single 500 mg dose (*FDA, 2020*). This is above the concentration needed to achieve antagonism in vitro and could prove problematic in patients. When we examined nafcillin-related molecules, we found that methicillin and oxacillin also antagonize FLZ. Furthermore, two cephalosporins often used in place of nafcillin, cefonicid and cefazolin, also unexpectedly antagonize fluconazole activity. Nafcillin has previously been shown to adversely affect patients on the drug warfarin due to nafcillin's induction of cytochrome P450 (*King et al., 2018*), which decreases warfarin concentration. This has been seen in a similar combination of FLZ and the antibiotic rifampicin (*Panomvana Na Ayudhya et al., 2004*). Rifampicin is a potent inducer of drug metabolism due to elevation of hepatic cytochrome P450 through increased gene expression (*Bolt, 2004*). This combination did indeed lower the levels of FLZ (*Panomvana Na Ayudhya et al., 2004*), resulting in relapse of cryptococcal meningitis (*Coker et al., 1990*). We hypothesize that an analogous process occurs when nafcillin is combined with FLZ, with nafcillin increasing ergosterol biosynthesis enzymes, counteracting FLZ's activity. In a recent autopsy study, 10 or 16 patients who died of cryptococcosis were administered either a penicillin or a cephalosporin (*Hurtado et al., 2019*). We recommend that these patients receive linezolid or vancomycin instead, since these drugs are used for similar bacterial targets but do not antagonize fluconazole activity (*Figure 5N*).

Our data demonstrate that O2M identifies promising new antifungal treatments that can rapidly move into the clinic. Our new combination of dicyclomine + FLZ almost doubled the median time-to-endpoint of mice treated with human dosages of dicyclomine (*Figure 7*), which is lower than dicyclomine's fungal MIC. Additionally, dicyclomine + FLZ appear to kill *C. neoformans* cells, not just inhibit *C. neoformans* cell growth, which increases this combination's therapeutic potential (*Figure 6M* and *Figure 6—figure supplement 2*). Dicyclomine's serum concentration in mice administered 60 μg was 0.2–0.6 μg/mL at 18 hr post injection (*Karaka et al., 2004*). While this is lower than dicyclomine's antifungal MIC, dicyclomine improves mouse survival when combined with FLZ even at these low doses (*Figure 7*). Dicyclomine is orally bioavailable and able to cross the blood brain barrier (*Das et al., 2013*; *Koerselman et al., 1999*), which makes it particularly promising for fungal meningitis treatment. Since dicyclomine, like many of our new FLZ synergizers, is approved by the FDA for other indications, it could rapidly move into the clinic.

In sum, O2M considerably streamlined the identification of important drug interactions affecting *C. neoformans* growth. These interactions are both synergistic and antagonistic among multiple fungal species capable of causing disease in humans. We focused on FDA-approved molecules to bypass the time and considerable expense it takes to develop a new drug (*Pushpakom et al., 2019*). However, our method would work equally well on any library of small molecules or biologic drugs to discover new antifungals. We showed that identifying these drug interactions can quickly lead to additional interacting pairs by examining structure (*Figure 3*) or by investigating underlying mechanism (*Wambaugh et al., 2017*). Finally, our newly discovered interaction of dicyclomine and FLZ exhibited therapeutic potential in vivo, demonstrating the clinical potential of fluconazole-containing synergistic pairs in the clinic.

## Materials and methods

### Fungal strains

Screening, validation, and structurally similar assays were performed with CM18 lab strain of *C. neoformans*. Screening with synergy prediction mutants (*CNAG_00573Δ* and *CNAG_03917Δ*) was in the CM18 background. Mechanistic studies were tested using the KN99 lab strain of *C. neoformans*. Clinical and environmental isolates of *C. neoformans* tested were a gift from Dr. John R. Perfect. *C. deuterogattii* strain R265 was purchased from ATCC. *Candida albicans* reference strain SC5314 and *Candida glabrata* reference strain CBS138 were used. *Candida auris* strains AR0383 and AR0384 were from the CDC.

### Microsource spectrum library screen

We inoculated either CM18 wild-type or *CNAG_00573Δ* or *CNAG_03917Δ* cells at 1000 cells per well of YNB + 2% glucose, then added small molecule to a final concentration of 1 μM. Plates were incubated for 48 hr at 30°C. $OD_{600}$ was measured on the BioTek plate reader model Synergy H1. Small molecules that altered growth by absolute value 0.22 in both synergy prediction mutants but not the wild type strain was considered significant. We found altered growth by 0.22 to give us the lowest false discovery rate when testing known synergistic and non-synergistic molecules.

### *C. neoformans* growth and small-molecule assays

All assays were performed in 1x YNB + 2% glucose. To determine MICs, an overnight culture was grown at 30°C with rotation, diluted to $OD_{600}$ = 0.02925 and 1000 cells were added to each well (2 μL of culture into 100 μL of media per well). Plates were incubated at 30°C unless otherwise stated. Small molecules were dissolved in DMSO to their highest soluble concentration and gradients were diluted in 2-fold dilution series. MIC values were calculated after 48 hr of incubation below the inhibitory effects of DMSO alone.

### Checkerboard assay and FICI calculations

We followed previously published methods (*Hsieh et al., 1993*; *Orhan et al., 2005*). Starting inoculation of either fungal strain was 2 μL of an $OD_{600}$ = 0.02925 (about 1000 cells per well of 100 μL medium). This inoculum was used for all fungal species and strains. Standard 96-well plates were grown statically for 48 hr at 30°C with minor shaking/resuspension of cells at 24 and 48 hr. Checkerboards were read at 0 and 48 hr on a BioTek plate reader model Synergy H1 to measure the $OD_{600}$ (*Candida albicans* and *Candida glabrata* were read at 0, 24, and 48 hr). Growth inhibition was assessed and FICIs for 50% and 90% inhibition were determined using standard methods (see *Wambaugh and Brown, 2018*). Briefly, to determine the FICI for each tested drug combination, we use the following formula to calculate FICI for each well in a 96 well plate that exhibits an $OD_{600}$ of ≤90% of the no drug control well: FICI = (concentration of drug #1 in well)/($MIC^{90}$ of drug #1) + (concentration of drug #2 in well)/($MIC^{90}$ of drug #2). FICI ≤0.5 is considered synergistic and FICI ≥4.0 is considered antagonistic. When testing for a synergistic interaction, the FICI is determined by the lowest scoring well in the plate. The FICI-determining well must exhibit a 4-fold decrease in drug concentration for each drug compared to the $MIC^{90}$ of each drug alone. For antagonistic interactions, the FICI is determined by the highest scoring well in a plate. Repeated results were averaged for the average FICI. Outliers with a different result (e.g. All replicates were performed on different days from independent stating cultures and are independent biological replicates. FICI scores presented in the figures are the average of a minimum of two independent replicates. If two replicates do not yield the identical result (i.e. both synergistic or neither synergistic), we repeat the assay and average the FICI scores to produce the final score. Outlier FICI scores are not included in the analysis because FICI scores are calculated on a $log_2$ scale and a single outlier can skew the results. Outlier scores are defined as those that differ from the majority of scores (e.g. if four replicates yielded FICI score of 0.25, 0.31125, 0.5, and 1.0, the 1.0 score is the only non-synergistic result of the four and would be considered an outlier). However, average these example scores would result in an FICI of 0.515, which would not be considered synergistic. As this average differs from the result of the majority of the FICI scores, the outlier is excluded from the calculation.

## Bliss independence assay

We created a gradient of fluconazole in a 96-well plate, then added small molecules at 10 µM or 100 nM final concentrations or vehicle. CM18 wild-type was added at 1000 cells per well of YNB+ 2% glucose. Percent growth was calculated for fluconazole, combinations, or small molecules alone. We then determined if growth inhibition caused by the combination was equal or greater than growth inhibition of the small molecules alone. Repeated results were averaged for the average Bliss score. All replicates were performed on different days from independent stating cultures and are independent biological replicates. Bliss scores presented in the figures are the average of a minimum of two independent replicates. If two replicates do not yield the identical result (i.e. both negative indicated synergy or both positive indicating not interaction), we repeated the assay and average the Bliss scores to produce the final score. Outlier Bliss scores are not included in the analysis they can skew the results. Outlier scores are defined as those that differ from the majority of scores (e.g. if four replicates yielded bliss scores < 0 and one score dramatically >0, the >0 score is the only non-synergistic result and would be considered an outlier). A DMSO control was conducted each time a molecule was tested to ensure the assay was working correctly. All DMSO results were averaged for the final score.

## Sterol extraction

KN99 culture was grown overnight in YNB + 2% glucose. Cells were sub-cultured into various treatments (vehicle control was YNB + 2% glucose). 6 ODs of each culture were harvested and lyophilized overnight. Pellets were resuspended in 25% alcoholic potassium hydroxide, vortexed, and incubated at 85°C water bath for 1 hr. Water and *n*-heptane were added to each tube, vortexed, and the *n*-heptane layer was transferred to borosilicate glass tubes. Biological replicates were grown on separate days from independent starting cultures.

## Metabolomics

Metabolomics analysis was performed at the Metabolomics Core Facility at the University of Utah which is supported by 1 S10 OD016232-01, 1 S10 OD021505-01, and 1 U54 DK110858-01.

## Resistance to dicyclomine screen

YNB + 2% glucose agar plates with or without 1.65 mg/mL of dicyclomine were made. Deletion mutants in KN99 strain were grown in YNB + 2% glucose then pinned to YNB plates. Plates were assessed at 1, 2, and 3 days for resistance.

## Cell permeability assay

An overnight culture of KN99 was grown in YNB + 2% glucose. This was then sub-cultured into the various treatments (vehicle control was either 0.1% DMSO or only YNB + 2% glucose). Cultures were grown at 30°C with rotation for 24 hr. Cultures were washed twice and resuspended in PBS. 3 µL of propidium iodide (stock concentration = 1 mg/mL) was added to the FACS tube. After 1 min, flow cytometry was performed. Voltage gates used were as follows: FSC: 500; SSC: 310; PE: 496. 100,000 cells were counted per sample. Experiments were repeated three times on separate days from a different starting culture for each experiment and thus represent biological replicates.

## Resistance to toxic amino acids

An overnight culture of KN99 was grown in YNB + 2% glucose. This was sub-cultured into either dicyclomine (0.3 mg/mL), FLZ (3E-4 mg/mL), Synergy, or Vehicle (YNB + 2% glucose) and grown at 30°C with rotation for 24 hr. Cells were then sub-cultured again into honeycomb plates with those previous treatments (dicyclomine, FLZ, Synergy, or Vehicle) with either 20, 10, 5, 2.5 ug/mL of 5-FAA or 0.4 mg/mL 5-MT or vehicle (3.2% DMSO). Plates were incubated at at 30°C in a Bioscreen C (Growth Curves USA) which automatically takes $OD_{600}$ measurements every 30 min.

## Calcofluor white flow cytometry

An overnight culture of KN99 was grown in YNB + 2% glucose then sub-cultured into either dicyclomine, FLZ, Synergy, or Vehicle (YNB + 2% glucose) and grown at 30°C with rotation for 24 hr. Cells were washed with PBS and calcofluor white added to a final concentration of 50 µg/mL and stained

for 5–15 min. Cells were washed once more with PBS and assessed by flow cytometry. Voltage gates as follows: FSC: 500; SSC: 317; BV421-A: 185. Significance determined with Mann-Whitney test.

## Fungicidal assay

This assay was adapted from *McMullan et al. (2015)*. KN99 was grown on YPAD agar plates and resuspended in YNB + 2% glucose. Cells were counted and transferred for a final concentration of 3 $\times 10^6$ cells/mL cultures containing treatments. Vehicle was YNB + 2% glucose. Cultures were left grown at 30° C statically for 3, 9, or 24 hr. Heat killed cultures were placed in a water bath >65° C for at least one hour. Cultures were then spun down and resuspended in PBS containing BCECF-AM (Invitrogen) at a final concentration of 40 ug/mL. Cultures were stained for 15 min in the dark then washed and resuspended in PBS and assessed by flow cytometry. 50000 cells were collected for each sample. Each treatment was a combination of at least three independent experiments. Voltage gates as follows: FSC: 500, SSC: 250, FITC-A: 230. Cells were gated on single cells then for BCECF-AM staining. Significance was determined with either uncorrected Fisher's Least Significant Distance test or Mann-Whitney test. Counts of CFU following heat killing of 3 $\times 10^6$ cells/mL are shown in *Figure 6—source data 1*.

## Murine model

~8 week-old female CD-1 IGS outbred mice (Charles River Laboratories) were intranasally inoculated with *C. neoformans* strain KN99 and monitored for survival as follows. The inoculum was prepared by culturing *C. neoformans* overnight in liquid YNB+2% glucose medium. *C. neoformans* cells collected from the overnight culture were washed twice in 1XPBS, counted, and suspended at a concentration of 5 $\times 10^5$ cells / mL. Mice were anesthetized with ketamine/dexmedetomidine hydrochloride (Dexdomitor) administered intraperitoneally (i.p.) and suspended by their front incisors on a line of thread. The inoculum was delivered intranasally by pipetting 50 μL (2.5 $\times 10^4$ cells / mouse) dropwise into the nostrils. After 10 min, mice were removed from the thread and were administered atipamezole (Antisedan) i.p. as a reversal agent. Mice were massed daily and euthanized by $CO_2$ asphyxiation when they reached 80% of their initial mass. Beginning 8 days post-inoculation, they received daily i.p. injections of either 8 mg/kg FLZ, 1.15 mg/kg DIC, 2.30 mg/kg DIC, 8 mg/kg FLZ + 1.15 mg/kg DIC, 8 mg/kg FLZ + 2.30 mg/kg DIC, or PBS control (vehicle). Dosages were determined from human doses (*Lexicomp, 2019a*; *Lexicomp, 2019b*). All animal studies were approved by the Institutional Animal Care and Use Committee at the University of Utah.

## Acknowledgements

We thank the University of Utah Metabolomics Core for sterol quantification, Jerry Kaplan, Ph.D. for advice on sterol extraction, and members of the Brown and Mulvey labs for helpful discussion and feedback. Fluconazole-resistant fungal strains were a gift from Dr. Kimberley Hanson at ARUP Laboratory. This work was supported by NIH grant R01AI137331 and funds from the Pathology Department at the University of Utah to JCSB. STD is supported by T32AI055434.

## Additional information

### Funding

| Funder | Grant reference number | Author |
| --- | --- | --- |
| National Institutes of Health | R01AI137331 | Jessica C S Brown |
| National Institutes of Health | T32AI055434 | Steven T Denham |

The funders had no role in study design, data collection and interpretation, or the decision to submit the work for publication.

### Author contributions

Morgan A Wambaugh, Conceptualization, Data curation, Formal analysis, Supervision, Validation, Investigation, Visualization, Methodology, Writing - original draft, Writing - review and editing;

Steven T Denham, Supervision, Investigation, Methodology, Writing - review and editing; Magali Ayala, Brianna Brammer, Miekan A Stonhill, Investigation; Jessica CS Brown, Conceptualization, Formal analysis, Supervision, Funding acquisition, Visualization, Project administration, Writing - review and editing

**Author ORCIDs**
Morgan A Wambaugh (iD) https://orcid.org/0000-0001-6663-2017
Jessica CS Brown (iD) https://orcid.org/0000-0002-3255-1486

**Ethics**
Animal experimentation: This study was performed in strict accordance with the the Guide for the Care and Use of Laboratory Animals of the National Institutes of Health. All of the animals were handled according to approved institutional animal care and use committee (IACUC) protocols (#18-006) of the University of Utah.

**Decision letter and Author response**
Decision letter https://doi.org/10.7554/eLife.54160.sa1
Author response https://doi.org/10.7554/eLife.54160.sa2

# Additional files

**Supplementary files**
• Supplementary file 1. Small molecules predicted to synergize with fluconazole by O2M. Small molecules predicted to interact with FLZ. Bioactivity and Status determined by Microsource Spectrum Library. The specific manufacturers we purchased molecules from are listed in last column. Molecules we did not purchase (for various reasons), have the manufacturer listed as N/A. INN, International Nonproprietary Names; USAN, United States Accepted Name; BAN, British Approved Names; JAN, Japanese Adopted Name; USP, United States Pharmacopeia; NF, National Formulary.

• Supplementary file 2. Minimum inhibitory concentrations of non-interacting molecules. Small molecules predicted to interact with FLZ but had no interaction. Minimum inhibitory concentration for 50% inhibition (MIC 50) and 90% inhibition (MIC 90) of small molecules predicted to interact with FLZ but resulted in no interaction. All values are against *C. neoformans* strain CM18. Bioactivity determined by Microsource Spectrum Library.

• Supplementary file 3. Minimum inhibitory concentrations of general anti-*C. neoformans* molecules.

• Supplementary file 4. Additional fungal species and strains used. Strains and strain sources used in this study.

• Supplementary file 5. Minimum inhibitory concentrations for various fungal strains/species. Minimum inhibitory concentration for 50% inhibition (MIC 50) and 90% inhibition (MIC 90) of small molecules that interacted with FLZ in various fungal species. N/A represents molecules that did not have an MIC.

• Supplementary file 6. Minimum inhibitory concentrations for FLZ resistant strains and species. Minimum inhibitory concentrations for 50% inhibition (MIC 50) and 90% inhibition (MIC 90) of FLZ resistant species. N/A represents molecules that did not have an MIC.

• Supplementary file 7. Gene deletion mutants resistant to Dicyclomine. Gene knockouts in KN99 resistant to dicyclomine at 1.65 mg/mL.

• Transparent reporting form

**Data availability**
All data generated or analyzed during this study are included in the manuscript and supporting files.

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
