## [Decision Letter]

**Acceptance summary:**

More than a million people, primarily with immunocompromised immune systems, die each year from invasive fungal infections, yet only a handful of antifungal drugs are currently available. To address this problem, the authors used a high throughput method to identify drugs (including many that are already FDA approved) that interact either synergistically or antagonistically with fluconazole, a major, widely available, and multi-species active antifungal drug but one which requires long periods of treatment. The authors identified drugs that act synergistically with fluconazole as well as drugs that act antagonistically with it. Although most of the testing was performed against the major pathogen *Cryptococcus neoformans*, the authors show that these antagonistic/synergistic interactions with fluconazole hold true for diverse fungal pathogens. The identification and characterization of drug interactions for treating diverse types of fungal infections harbors substantial potential for improving patient outcomes.

**Decision letter after peer review:**

Thank you for submitting your article "Synergistic and Antagonistic Drug Interactions in the Treatment of Systemic Fungal Infections" for consideration by *eLife*. Your article has been reviewed by two peer reviewers, and the evaluation has been overseen by a Reviewing Editor and Wendy Garrett as the Senior Editor. The reviewers have opted to remain anonymous.

The reviewers have discussed the reviews with one another and the Reviewing Editor has drafted this decision to help you prepare a revised submission.

Essential revisions:

Thank you for submitting this interesting manuscript to *eLife*. All of us thought that the data is extensive and the results are conclusive and that the manuscript is potentially worthy of publication. However, all of us also thought that the manuscript needs some additional work, especially in a) clearly explaining what was done and why, and b) interpreting and discussing the results. Specifically:

1) While this manuscript is well written, the second part of the paper is difficult to follow as it lacks focus and a clear rationale. The part starting with the identification of general anti-cryptococcal molecules by O2M and everything following appears to combine different angles of thought that are presented without a unifying thread.

2) Understanding O2M based on the information provided in the Introduction is difficult and relies too heavily on knowledge from two previous publications from this group. It would greatly help the reader of this manuscript if the method would be clearly described as to stand alone.

3) Subsection “Synergy prediction mutants for fluconazole allow for high-throughput screening of small 117 molecule interactions”, first paragraph: Similarities between chemical-genetic signatures appear to be a key part of the O2M concept. Please explain how similar they have to be and how similarity was assessed.

4) The subsection “Bliss independence Assay” states that the 'most repeatable result' was averaged for a Bliss independence score. Please explain, how many and which ones were averaged, given that there are seven replicates for DMSO for example and some drugs have a mix of positive and negative scores.

5) The screen is based on an assay that identifies two or more gene deletion strains whose growth is affected by two known synergistic drugs. Those gene deletion strains are then used to screen for other new drugs that have the same effect on the gene deletions. Points to be considered:

a) The gene deletions used are identified by gene number, but not by name. It would be useful to a reader to know the genes are a hypothetical gene and a nuclear pore gene. The significance of these genes is not obvious.

b) It is clear that the screen is highly successful. However, the screen suggests that the synergy uses the same mechanisms as the known drug synergy. Would not synergy by other mechanisms not affect the same two gene deletions and therefore not be detected in the screen. This should be discussed.

6) Table 1 – points to be considered: a) It would be more useful if sorted by results, and then alphabetical.

b) The text does not mention drugs that have an MIC50 but not an MIC90 – presumably this is transient growth or trailing, which has been described in Ca and Cn for many years. The manuscript needs to describe and discuss this.

c) It is not clear how a strain has an MIC90 but not an MIC50 – why is it not available. It might be the same as the MIC90 – that might be expected. It might be less than the first drug concentration used. That might be expected. But it should be available, not NA. MIC50 N/A appears in many tables and needs to be corrected.

d) The MIC are reported to three significant digits which is commendable if true. Presumably these are real drug concentrations tested, not an extrapolation of tested drug concentrations. Three significant digits for a drug concentration is somewhat unrealistic.

e) It would be useful throughout the text to know what serum concentration can be achieved in the blood for these FDA approved drugs. That information should be easily obtained and would provide a check on the likely use of these drugs in humans.

7) Figure 1B: a) The legend should state that the red part of a chromosome indicates the gene deletion (if that is what it is?).

b) The yeast cell dotted outline is meant to indicate lack of cell growth? It should be stated in the legend.

c) The X axis on the FICI assay should be labeled, possibly "Identified molecules".

8) Figure 2C would benefit from a log scale, not a split scale. (possible both Figure 2C and 2A could use the same scale).

9) Figure 3 – this is the most problematic figure in the paper: a) The X axis for each panel is different so a casual reader can easily be confused. The same scale should be used for each panel.

b) The Y axis should indicate the species for the first 12 isolates, perhaps as "Cn".

c) The drugs that show synergy (green) should be in one section of the figure and the drugs that show antagonism (purple) should be in another section. It is possible that the green drugs and purple drugs could have different X axis scales as long as they are all the same within each of the two colors.

10) Figure 4: a) It is unlikely that the casual reader will appreciate the drug structures. This might best to put into the supplementary figures.

b) Figure 4N lists the drugs but is not linked to the structures. If the structures stay, then indicate which structure goes with which drug without having to find it in the legend.

c) Figure 4N – add nafcillin sodium to the data.

d) Figure 4P – Presumably the only significant differences to control are the two labeled as significant. Is the p value of 0.0268 for Figure 4P or Q? It is not usually to 3 significant digits. If that value is for Figure 4Q, what is the p value for Figure 4P?

e) Statements in the text about concentration dependence from Figure 4P are not supported and should be removed.

f) Figure 4Q can be stated as one sentence in the text. It should not be a figure panel.

11) Figures 5 and 6: a) There are two figures presented that seem to be almost the same. The text mentions Figure 6 with no mention of Figure 5. This should be resolved.

b) Figure 5I-L should be labeled in the figure with 5FAA.

c) The legend needs to state the concentrations of each of the drugs (dicyclomine and fluconazole), especially for synergy. Presumably the concentrations on the panels are for 5FAA.

d) Figure 6L uses a different scale from panels I to K, or to Figure 5I to L. The scale should not be changed. Figure 6L should not be used.

e) The legend or text should comment on the achievable concentration of dicyclomine in the mouse serum.

12) Supplementary file 1 – presumably the drugs labeled N/A for manufacturer are only available as part of the library. This should be made clearer.

13) Figure 2—figure supplement 1: a) How do you calculate a standard deviation from two independent experiments? Isn't that just the average?

b) The two sections of the figure should be labeled 10µM and 100 µM.

14) Supplementary Figure 3 should be in the body of the manuscript, as with Figure 4. Again, the structures could remain in the supplementary figures, but panel M should be in the body of the paper. Or the structures should be labeled to compare to the panel.

---

## [Author Response]

1) While this manuscript is well written, the second part of the paper is difficult to follow as it lacks focus and a clear rationale. The part starting with the identification of general anti-cryptococcal molecules by O2M and everything following appears to combine different angles of thought that are presented without a unifying thread.

Thank you for this helpful feedback. We have rearranged some sections to improve flow and rewritten others to improve the narrative and better connect the experimental goals.

2) Understanding O2M based on the information provided in the Introduction is difficult and relies too heavily on knowledge from two previous publications from this group. It would greatly help the reader of this manuscript if the method would be clearly described as to stand alone.

We have re-written this section (subsection “Synergy prediction mutants for fluconazole allow for high-throughput screening of small molecule interactions”) to clarify O2M.

3) Subsection “Synergy prediction mutants for fluconazole allow for high-throughput screening of small 117 molecule interactions”, first paragraph: Similarities between chemical-genetic signatures appear to be a key part of the O2M concept. Please explain how similar they have to be and how similarity was assessed.

There does not need to be much overlap between chemical-genetic signatures to identify synergy prediction genes. Our starting chemical-genetic signatures contained phenotypes from ~1400 knockout mutants and we identified three synergy prediction mutants. We clarified this in the text (subsection “Synergy prediction mutants for fluconazole allow for high-throughput screening of small molecule interactions”).

4) The subsection “Bliss independence Assay” states that the 'most repeatable result' was averaged for a Bliss independence score. Please explain, how many and which ones were averaged, given that there are seven replicates for DMSO for example and some drugs have a mix of positive and negative scores.

Each molecule was a result of a minimum of two experiments, though often tested in three independent experiments and any outlier result was excluded in the calculation. A DMSO control was conducted each time an experiment was and all DMSO results were averaged for the final control result. We clarified this in the text (subsection “Bliss independence Assay”).

5) The screen is based on an assay that identifies two or more gene deletion strains whose growth is affected by two known synergistic drugs. Those gene deletion strains are then used to screen for other new drugs that have the same effect on the gene deletions. Points to be considered:a) The gene deletions used are identified by gene number, but not by name. It would be useful to a reader to know the genes are a hypothetical gene and a nuclear pore gene. The significance of these genes is not obvious.

We added the predicted function (based on homology) to the text (Synergy prediction mutants for fluconazole allow for high-throughput screening of small molecule interactions). We use these genes as a tool, so the importance of their putative function is not known.

b) It is clear that the screen is highly successful. However, the screen suggests that the synergy uses the same mechanisms as the known drug synergy. Would not synergy by other mechanisms not affect the same two gene deletions and therefore not be detected in the screen. This should be discussed.

The reviewers raise a good point that the molecular mechanism of dicyclomine inhibition is similar to one of the proposed mechanisms for sertraline, one of our starting synergizers. This suggests some limitations for the types of synergistic drugs we could identify by O2M. Our work on antibiotics shows that there is some utility in this, as when we identified a synergistic pair that phenocopied a known synergizer but targeted different proteins, we found that the new synergistic pair could inhibit growth of *E. coli* strains resistant to the old pair. We address these points in the Discussion (first paragraph).

6) Table 1 – points to be considered: a) It would be more useful if sorted by results, and then alphabetical.

We made this change.

b) The text does not mention drugs that have an MIC50 but not an MIC90 – presumably this is transient growth or trailing, which has been described in Ca and Cn for many years. The manuscript needs to describe and discuss this.

All molecules were dissolved to their highest possible soluble concentration in DMSO. This soluble drug was added to YNB at a concentration below DMSO inhibitory effect (DMSO showed inhibitory effects around 5%). In some cases, solubility in aqueous solutions was too low to inhibit fungal growth. It is also possible that these are trailing growth. We did not test subsequent timepoint to determine if this MIC50 changed. We have added this information to the text (Synergy prediction mutants for fluconazole allow for high-throughput screening of small molecule interactions”).

c) It is not clear how a strain has an MIC90 but not an MIC50 – why is it not available. It might be the same as the MIC90 – that might be expected. It might be less than the first drug concentration used. That might be expected. But it should be available, not NA. MIC50 N/A appears in many tables and needs to be corrected.

The reviewers are correct: these were the molecules for which the MIC50 and MIC90 were the same. We have added these data to the table.

d) The MIC are reported to three significant digits which is commendable if true. Presumably these are real drug concentrations tested, not an extrapolation of tested drug concentrations. Three significant digits for a drug concentration is somewhat unrealistic.

We have corrected this.

e) It would be useful throughout the text to know what serum concentration can be achieved in the blood for these FDA approved drugs. That information should be easily obtained and would provide a check on the likely use of these drugs in humans.

We added serum concentration for dicyclomine (Discussion, sixth paragraph) and nafcillin (Discussion, fifth paragraph) to the text.

7) Figure 1B: a) The legend should state that the red part of a chromosome indicates the gene deletion (if that is what it is?)

The red part of the chromosome indicates gene deletion. We have added this to the figure legend.

b) The yeast cell dotted outline is meant to indicate lack of cell growth? It should be stated in the legend.

The dotted yeast cell indicates decreased cell growth. We have added this to the figure legend.

c) The X axis on the FICI assay should be labeled, possibly "Identified molecules"

We added this to the figure.

8) Figure 2C would benefit from a log scale, not a split scale. (possible both Figure 2C and 2A could use the same scale).

We have changed Figure 2C as suggested. Figure 2A was difficult to see on this scale, so we kept it the same.

9) Figure 3 – this is the most problematic figure in the paper:

Note: The original Figure 3 is now Figure 4.

a) The X axis for each panel is different so a casual reader can easily be confused. The same scale should be used for each panel.

This has been changed.

b) The Y axis should indicate the species for the first 12 isolates, perhaps as "Cn".

This has been added.

c) The drugs that show synergy (green) should be in one section of the figure and the drugs that show antagonism (purple) should be in another section. It is possible that the green drugs and purple drugs could have different X axis scales as long as they are all the same within each of the two colors.

This has been changed.

10) Figure 4:

Note: The original Figure 4 is now Figure 5.

a) It is unlikely that the casual reader will appreciate the drug structures. This might best to put into the supplementary figures.

We have chosen to keep the structures but labeled them better for the reader.

b) Figure 4N lists the drugs but is not linked to the structures. If the structures stay, then indicate which structure goes with which drug without having to find it in the legend.

We have indicated which structure is which drug. (Note: Figure 4N is now Figure 4M).

c) Figure 4N – add nafcillin sodium to the data.

This has been added.

d) Figure 4P – Presumably the only significant differences to control are the two labeled as significant. Is the p value of 0.0268 for Figure 4P or Q? It is not usually to 3 significant digits. If that value is for Figure 4Q, what is the p value for Figure 4P?

The p value is for Figure 4P. Figure 4Q is an FICI score which 0.5 and below is considered synergistic as previously described. We removed Figure 4Q from the figure and corrected the p value.

e) Statements in the text about concentration dependence from Figure 4P are not supported and should be removed.

We have made this change.

f) Figure 4Q can be stated as one sentence in the text. It should not be a figure panel.

We have changed this and taken it out of the figure.

11) Figure 5/6: a) There are two figures presented that seem to be almost the same. The text mentions Figure 6 with no mention of Figure 5. This should be resolved.

We apologize for this was a mistake. It has been resolved.

b) Figure 5I to L should be labeled in the figure with 5FAA.

We added this to the graphs.

c) The legend needs to state the concentrations of each of the drugs (dicyclomine and fluconazole), especially for synergy. Presumably the concentrations on the panels are for 5FAA.

We added this to the legend.

d) Figure 6L uses a different scale from Figure 6I to K, or to Figure 5I to L. The scale should not be changed. Figure 6L should not be used.

We apologize for this mistake. The graphs now have the same scale.

e) The legend or text should comment on the achievable concentration of dicyclomine in the mouse serum.

Previous studies show that dicyclomine concentrations in the sera of mice given a 60ug dose was 0.5-1.5 µg/mL at 0 hours and 0.2-0.6 µg/mL after 18 hours. We have added this information to the main text (Discussion, sixth paragraph).

12) Supplementary file 1 – presumably the drugs labeled N/A for manufacturer are only available as part of the library. This should be made clearer.

We added information to the legend.

13) Figure 2—figure supplement 1: a) How do you calculate a standard deviation from two independent experiments? Isn't that just the average?

For the bar graphs that represent only two data points, the standard deviation and the range are the same.

b) The two sections of the figure should be labeled 10 µM and 100 µM.

We have added this to the figure.

14) Supplementary Figure 3 should be in the body of the manuscript, as with Figure 4. Again, the structures could remain in the supplementary figures, but panel M should be in the body of the paper. Or the structures should be labeled to compare to the panel.

This is now Figure 3. We labeled the structures to facilitate comparison to the graph.